# Mutually attracting spin waves in the square-lattice quantum antiferromagnet

**Michael Powalski[1], Kai P. Schmidt[2⋆] and Götz S. Uhrig[1†]**

**1** Lehrstuhl für Theoretische Physik 1, TU Dortmund, Germany
**2** Lehrstuhl für Theoretische Physik I, Staudtstraße 7, Universität Erlangen-Nürnberg, D-91058 Erlangen, Germany

⋆ kai.phillip.schmidt@fau.de,  † goetz.uhrig@tu-dortmund.de

## Abstract

Spin waves (magnons) in two dimensions are the potential glue in high-temperature superconductors so that their quantitative understanding is mandatory. Yet even for the fundamental case of the undoped Heisenberg model on the square lattice a consistent picture is still lacking. Significant spectral continua are taken as evidence of the existence of fractional excitations (spinons), but descriptions in terms of spinons fail to show the established absence of an energy gap. Here a fully consistent picture of the dynamics in the square-lattice quantum antiferromagnet is provided which agrees with the experimental findings. The key step is to capture (i) the strong attractive interaction between the spin waves and (ii) the vertex corrections of the observables.


# 1 Introduction

The spin-$\frac{1}{2}$ Heisenberg model is one of the simplest and most paradigmatic models in quantum magnetism [1]. Its relevance has been boosted further by the discovery of cuprate high-temperature ($T_c$) superconductors [2] as their undoped parent compounds realize the Heisenberg model on the square-lattice [3]. Tremendous experimental and theoretical effort has been invested in studying their magnetic excitations which are believed to provide the glue of high-$T_c$ superconductivity.

Much is known about the elementary excitations at large wavelengths [4–7], described by spin waves (magnons), the Goldstone bosons of the long-range ordered antiferromagnetic phase [8]. But their nature at short wavelengths remains unclear to this day. Yet precisely the short-range processes play a decisive role in the understanding of high $T_c$ superconductivity [9–12].

Since the seminal paper by Anderson [13] there are wide-spread activities to seek for fractionalization of magnons into spinons [14–16]. By contrast, here we present strong evidence that spinons do not appear as the elementary excitations at any wavevector. We derive a comprehensive picture in terms of dressed magnons which agrees strikingly with experimental data. Their anomalous behavior at large wavevectors can be traced back to a strong mutual attraction on short-length scales as sketched in Fig. 1. Thus, our results provide key information on the nature of the magnetic excitations in the important class of parent compounds of high-$T_c$ superconductors.

The quantum antiferromagnet on the square lattice (QASQ) is a paradigmatic example of long-range ordered quantum phases and broken continuous symmetries in condensed matter [3, 17]. In a previous work [18], we derived a quantitative description of the magnon dispersion including the roton minimum at high energies. The key step was to take the full renormalization of the magnon dispersion and of the magnon-magnon interactions into account. Technically, this was achieved by a continuous similarity transformation (CST) which

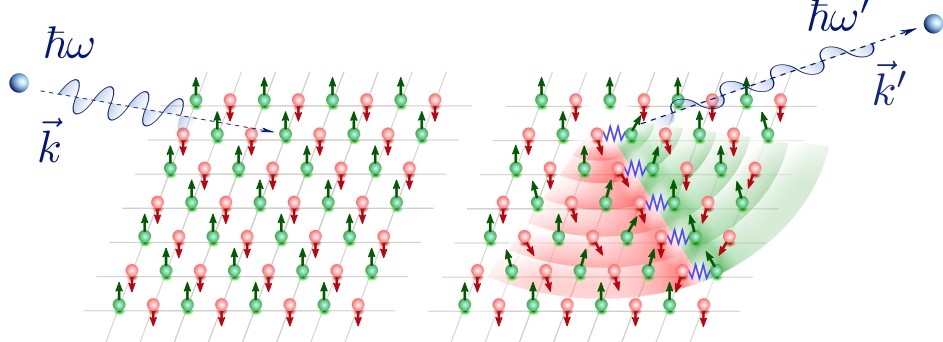

Figure 1: **Sketch of a scattering event.** By inelastic scattering of a neutron in the longitudinal channel two magnons are excited from the long-range ordered antiferromagnetic ground state. On short distances, they interact strongly and attract each other forming resonances [18]. Note that the magnons are sketched here as wave packets localized in *real* and in *reciprocal* space with a finite minimum uncertainty as required by Heisenberg's uncertainty principle.

was performed in a non-perturbative fashion, see also Sect. 3. This means that we changed the basis in which the model is described in such a way that the resulting effective model is easier to analyze. The attractive magnon-magnon interaction gives rise to the formation of a two-magnon "Higgs" resonance corresponding to a longitudinal magnon with finite lifetime. These calculations in the longitudinal channel were based on the bare observable, i.e., the observable was not transformed to the basis used for the Hamiltonian operator. This means that the vertex corrections were still lacking completely. Yet, they are absolutely crucial for the continuum in the transversal spin channel. Without them, no such continuum occurs. This vital piece of understanding of the inelastic scattering of polarized neutrons is added in the present work.

We aim at the quantitative description of static and dynamic correlations of single- and multiple-magnon states. This will allow us to compare the theoretical spectroscopic signatures of interacting magnons with recent experimental data. For this purpose we derive the effective observables which embody the vertex corrections. The systematic consideration of effective observables is a crucial extension of Ref. [18] because it allows us to analyze the spectroscopic features quantitatively. We emphasize that our approach does not require any resort to fractional spinons.

We show that the anomalous dispersion in the magnon spectrum is caused by substantial hybridization of single magnons with the three-magnon continuum. This hybridization is strongly enhanced by a strong attraction between pairs of magnons leading to significant continua. Our interpretation is supported by the noticeable agreement with recent experimental results [16].

The paper is organized as follows. In Sect. 2, we discuss the major intricacies which arise in the general magnon approach pointing out the relevance of magnon-magnon interactions. Section 3 renders a brief overview of the technical approach based on continuous transformations in momentum space. In Sect. 4, we present the spectroscopic properties of interacting magnons and compare the theoretical results with experimental data from inelastic scattering with polarized neutrons. Our results are interpreted theoretically in Sect. 5. We identify the origin of the increased weight in the high-energy spin wave continuum and discuss the relevance of mutual magnon attraction. The final Sect. 6 provides the conclusions and an outlook.

## 2   Renormalized magnon description

There is no exact solution of the QASQ, but its low-energy physics is well understood. There is overwhelming evidence that the ground state exhibits long-range Néel order for zero temperature [3]. This ordering is associated with a finite staggered magnetization which spontaneously breaks the continuous SU(2) symmetry of the Hamiltonian. As a result, the QASQ displays gapless bosonic excitations in accordance with Goldstone's theorem [8]. The corresponding bosons are quantized spin waves, i.e., magnons, which exhibit a linear dispersion at long wavelengths [19, 20]. Even in the case $S = 1/2$, where quantum fluctuations are strongest [6], magnons provide a quantitative and consistent description of the low-energy excitations. But their nature at high energies, i.e., at short wavelengths, is still not clarified to this day.

In the following, we give a brief sketch of the general spin wave formalism for the QASQ focusing on the physical interpretation and the intricacies of this approach. Details can be found in various standard textbooks [21, 22], reviews [3, 23], or in the original papers [19, 20, 24–27]. We consider the Heisenberg model with nearest-neighbor antiferromagnetic exchange interaction $J > 0$ between spins with $S = 1/2$

$$H = J \sum_{\langle i,j \rangle} \vec{S}_i \cdot \vec{S}_j \; . \tag{1}$$

At zero temperature the ground state exhibits long-range Néel order implying that the spin orientations favor an anti-parallel alignment on sublattice A and B of the bipartite square lattice. The classical Néel state $|\mathrm{AF}\rangle$ with ↑ spins on A and ↓ spins on B is chosen as a reference state of the antiferromagnetic system. The main idea of spin wave theory is to represent the deviations from this classical Néel state by bosonic excitations. To this end, spin operators are expressed by local bosonic creation and annihilation operators $a_i^{(\dagger)}$ and $b_j^{(\dagger)}$ discriminating between bosons on sublattice $A$ and $B$.

Here we choose the Dyson-Maleev representation [23–25] defined by the following relations

$$S_i^+ = \sqrt{2S} \left( a_i - \frac{a_i^\dagger a_i a_i}{2S} \right), \qquad S_i^- = \sqrt{2S} \, a_i^\dagger \; , \tag{2a}$$

$$S_j^+ = \sqrt{2S} \left( b_j^\dagger - \frac{b_j^\dagger b_j^\dagger b_j}{2S} \right), \qquad S_j^- = \sqrt{2S} \, b_j \; , \tag{2b}$$

$$S_i^z = S - a_i^\dagger a_i \qquad S_j^z = -S + b_j^\dagger b_j \; . \tag{2c}$$

The resulting Hamiltonian is equivalent to coupled harmonic oscillators with additional anharmonic interactions. The essential advantage of the Dyson-Maleev representation is that the interactions can be expressed by a finite number of *quartic* boson operators at the expense of the manifest hermitecity of the Hamiltonian. By contrast, a Holstein-Primakoff representation generally requires an expansion in $1/S$ due to the occurring square root expressions [6, 28].

We stress that the Dyson-Maleev representation is a faithful representation of the spin algebra. This means that the dynamical processes expressed by functions of the spins are faithfully expressed by the bosonic expressions (2) if they start from a physical state and end at a physical state, i.e., states with at maximum 2S bosons per site. In the literature [23, 24, 27] the distinction between dynamical and kinematical interactions is made. The dynamical part is the one which expresses multi-particle processes (quartic and possibly higher terms in the boson operators) in the Hamiltonian. The kinematical part results in the condition to include only physical states, i.e., states with at maximum 2S spins per site. We find this distinction

slightly misleading because the faithful representation of the spin algebra ensures already that unphysical states are not reached from physical states or do not lead to physical states. Thus, the essential aspects are covered by the "dynamical" interaction. This is the basis for the rigorous perturbative analyses of the spin dispersion and spin correlations [29–33]. Only at finite temperatures, the kinematic constraint has to be imposed additionally because otherwise unphysical states would contribute to the density operators.

Since we focus here on zero temperature response functions we henceforth only need the quartic terms ensuing from the Dyson-Maleev representation (2) and we will call them interactions if they consist of two incoming and two outgoing bosons. Otherwise, we call them hybridizations because they link one boson to three bosons. The attribute "dynamical" is omitted because the spin algebra ensures the kinematical aspects as well at zero temperature.

The bilinear bosonic terms of $H$ can be diagonalized by a Bogoliubov transformation in momentum space which introduces transformed bosons $\alpha_{\mathbf{k}}^{\dagger} = l_{\mathbf{k}} a_{\mathbf{k}}^{\dagger} + m_{\mathbf{k}} \beta_{-\mathbf{k}}$ and $\beta_{\mathbf{k}}^{\dagger} = m_{\mathbf{k}} a_{-\mathbf{k}} + l_{\mathbf{k}} b_{\mathbf{k}}^{\dagger}$ where we used the Fourier transformed boson operators

$$a_{\mathbf{k}}^{\dagger} = \frac{1}{\sqrt{N}} \sum_{\mathbf{r}_i \in A} a_i^{\dagger} e^{-i\mathbf{k}\cdot\mathbf{r}_j} , \qquad b_{\mathbf{k}}^{\dagger} = \frac{1}{\sqrt{N}} \sum_{\mathbf{r}_j \in B} b_j^{\dagger} e^{-i\mathbf{k}\cdot\mathbf{r}_j} , \tag{3}$$

with $N$ denoting the number of sites of the $A$ or the $B$ sublattice. We are working throughout this article with operators in the Hamiltonian in the magnetic Brillouin zone (MBZ) excluding the region around $\mathbf{k} = (\pi, \pi)$. Hence, the two gapless modes are found only at the origin while the dispersion, see below, is finite at the magnetic zone boundary. Note, however, that operators appearing in the structure factors and representing external (de)excitations can have momenta in the full Brillouin zone.

The explicit coefficients $l_{\mathbf{k}}$ and $m_{\mathbf{k}}$ are given in Ref. [34]. The resulting spin wave Hamiltonian can be written in the following form

$$H_{\text{init}} = H_{\text{SW}} + V^0 + V^+ + V^- \quad , \tag{4}$$

where the bilinear part

$$H_{\text{SW}} = E_0 + \sum_{\mathbf{k}} \omega_{\mathbf{k}} \left( \alpha_{\mathbf{k}}^{\dagger} \alpha_{\mathbf{k}} + \beta_{\mathbf{k}}^{\dagger} \beta_{\mathbf{k}} \right) \tag{5}$$

describes decoupled magnons modes. The coefficients

$$e_0^{(1)} := E_0/N = -2J(S^2 + AS + A^2/4), \tag{6a}$$

$$\omega_{\mathbf{k}} = 2J(2S + A)\sqrt{1 - \gamma(\mathbf{k})^2}, \tag{6b}$$

$$=: (1 + A/(2S))\omega_{\mathbf{k}}^{(0)}, \tag{6c}$$

$$A := \frac{2}{N} \sum_{\mathbf{k}} \left( 1 - \sqrt{1 - \gamma(\mathbf{k})^2} \right) \approx 0.157947, \tag{6d}$$

correspond to the ground-state energy per site and the one-magnon dispersion in next-to-leading order spin wave theory [29, 34].

The remaining non-diagonal part can be decomposed into quartic interaction terms

$$V^0 = \sum_{1234} \delta_{34}^{12} \left\{ V_{1234}^{(\alpha\alpha)} \alpha_1^{\dagger} \alpha_2^{\dagger} \alpha_3 \alpha_4 + V_{1234}^{(\beta\beta)} \beta_{-4}^{\dagger} \beta_{-3}^{\dagger} \beta_{-2} \beta_{-1} + V_{1234}^{(\alpha\beta)} \alpha_1^{\dagger} \alpha_3 \beta_{-4}^{\dagger} \beta_{-2} \right\}, \tag{7}$$

which conserve the number of magnons and into quartic hybridization processes

$$V^+ = \sum_{1234} \delta_{34}^{12} \left\{ V_{1234}^{(3-\alpha)} \alpha_1^{\dagger} \alpha_2^{\dagger} \beta_{-3}^{\dagger} \alpha_4 + V_{1234}^{(3-\beta)} \alpha_2^{\dagger} \beta_{-3}^{\dagger} \beta_{-4}^{\dagger} \beta_{-1} + V_{1234}^{(+4)} \alpha_1^{\dagger} \alpha_2^{\dagger} \beta_{-3}^{\dagger} \beta_{-4}^{\dagger} \right\}, \tag{8a}$$

$$V^- = \sum_{1234} \delta_{34}^{12} \left\{ V_{1234}^{(\alpha-3)} \alpha_1^{\dagger} \alpha_3 \alpha_4 \beta_{-2} + V_{1234}^{(\beta-3)} \alpha_3 \beta_{-4}^{\dagger} \beta_{-1} \beta_{-2} + V_{1234}^{(-4)} \alpha_3 \alpha_4 \beta_{-1} \beta_{-2} \right\}, \tag{8b}$$

which increase (+) or decrease (−) the number of magnons. Note that the changes of the number of magnons are *even* as a consequence of the collinear Néel order and the bipartiteness of the square lattice. Either two or four magnons are created in $V^+$ or annihilated in $V^-$. We use a shorthand notation for the momenta $\mathbf{k}_i \to \mathbf{i}$ such that $a_{\mathbf{k}_i}^\dagger := a_{\mathbf{i}}^\dagger$ to lighten the notation. The Kronecker symbol $\delta_{34}^{12}$ equals unity if $\mathbf{1} + \mathbf{2} = \mathbf{3} + \mathbf{4}$ modulo a reciprocal lattice vector and zero otherwise ensuring the conservation of total crystal momentum. The explicit vertex functions are given in Ref. [34].

The interaction and hybridization processes are a consequence of the algebraic properties of the spin operators represented by boson operators. They include the constraint of finite dimensional local Hilbert spaces if processes starting and ending at physical states are considered. The hybridization of single magnons with three-magnon states expressed in $V^+$ and $V^-$ turns the system into an intricate many-body problem. The interaction and the hybridization terms are proportional to $1/S$ as is obvious from the Dyson-Maleev representation (2). Thus, on the one hand, it is to be expected that their effect is particularly strong for $S = 1/2$. On the other hand, one finds that long-wavelength magnons propagate almost freely because the deviation from the Néel state is distributed in real space and the scattering due to the interaction is relatively small [24, 26, 27]. This is derived by mapping the microscopic model in the long-wavelength limit to a continuum model and analyzing it by renormalization group techniques [5]. Thus, the effect of the interaction terms is marginal in the limit of low energies and long wavelengths which is also reflected by the good accuracy of the conventional next-to-leading order spin wave theory where one neglects the terms $V^\pm$ and $V^0$. This approximation is essentially based on the assumption that $\langle a_i^\dagger a_i \rangle, \langle b_j^\dagger b_j \rangle \ll S$.

But in the case of magnons at short wavelengths, the interactions (7) as well as the hybridizing terms (8) can play a role in spite of their short range in real space. The short-wavelength magnons have high energies and thus their scattering and hybridizing processes dispose of a much larger phase space. A simplistic visualization of this fact is that magnons at short wavelengths can form much better localized wave packets at given relative uncertainty $\Delta k / k$ than magnons at long wavelengths. As a result, the conventional perturbative expansion in powers of $1/S$ turns out to be inefficient displaying only a slow convergence at $\mathbf{k} = (\pi, 0)$ as reported in Ref. [31]. We showed that the anomalous dispersion of the high-energy magnons at the zone boundary of the MBZ can be attributed to strong quantum effects caused by the interplay of magnon attraction and the hybridization terms at short wavelengths [18]. Consequently, the deviations of spin wave theory at the zone boundary are rather a result of a methodical insufficiency than indicating fundamentally different physics. The non-perturbative nature of high-energy magnons requires a sophisticated methodical treatment which goes beyond the conventional perturbative expansion in $1/S$.

## 2.1 Effective magnonic Hamilton operator

The fundamental idea of our approach is to map the initial Hamilton operator to an effective Hamilton operator in terms of magnons

$$\mathcal{H}_{\text{eff}} = \mathcal{H}_{\text{M}} + \mathcal{V}^0, \tag{9}$$

which conserves the number of magnons because no hybridization terms $\mathcal{V}^\pm$ appear anymore. The calligraphic $\mathcal{V}$ is used for the *renormalized* interaction after the change of basis. The initial quartic terms were denoted by straight $V$. The first part

$$\mathcal{H}_{\text{M}} = \mathcal{E}_0 + \sum_{\mathbf{k}} \omega_{\mathbf{k}}^{\text{eff}} \left( \alpha_{\mathbf{k}}^\dagger \alpha_{\mathbf{k}} + \beta_{\mathbf{k}}^\dagger \beta_{\mathbf{k}} \right) \tag{10}$$

describes renormalized magnons. The second part denotes the effective interactions

$$\mathscr{V}^0 = \sum_{1234} \delta_{34}^{12} \left\{ \mathscr{V}_{1234}^{(\alpha\alpha)} \alpha_1^\dagger \alpha_2^\dagger \alpha_3 \alpha_4 + \mathscr{V}_{1234}^{(\beta\beta)} \beta_{-4}^\dagger \beta_{-3}^\dagger \beta_{-2} \beta_{-1} + \mathscr{V}_{1234}^{(\alpha\beta)} \alpha_1^\dagger \alpha_3 \beta_{-4}^\dagger \beta_{-2} \right\} . \tag{11}$$

We stress that in the continuous transformation the effect of the hybridization processes in (4) are absorbed into the renormalized coefficients, i.e., into the effective ground state energy $\mathscr{E}_0$, the magnon dispersion $\omega_{\mathbf{k}}^{\text{eff}}$, and the effective interaction $\mathscr{V}^0$. In this way, the hybridization processes are accounted for by the renormalized Hamiltonian $\mathscr{H}_{\text{eff}}$ of the dressed magnons which constitute the true elementary excitations of the system.

## 2.2 Spectral densities

Spectral densities provide the theoretical description of the momentum and frequency resolved counting rates $I_{\text{exp}}(\omega, \mathbf{Q})$ measured in INS experiments. For sufficiently low temperatures the rate $I_{\text{exp}}(\omega, \mathbf{Q})$ is proportional to the dynamic structure factor (DSF) at zero temperature

$$S^{\alpha\alpha}(\omega, \mathbf{Q}) = -\frac{1}{\pi} \text{Im} \langle 0| S_{\text{eff}}^\alpha(-\mathbf{Q}) \frac{1}{\omega - (\mathscr{H}_{\text{eff}} - \mathscr{E}_0)} S_{\text{eff}}^\alpha(\mathbf{Q})|0\rangle , \tag{12}$$

where $|0\rangle$ is the ground state of the effective Hamiltonian $\mathscr{H}_{\text{eff}}$ with the renormalized ground state energy $\mathscr{E}_0$ and the $S_{\text{eff}}^\alpha(\mathbf{Q})$ are the Fourier transformed components of the effective spin operator with $\alpha = x, y, z$.

Let us assume that the staggered magnetization of the antiferromagnetic phase, which breaks the spin-rotation symmetry spontaneously, is aligned along the $z$-direction. Inelastic scattering experiments with polarized neutrons allow one to distinguish between the longitudinal part of the DSF for $\alpha = z$ and its transversal parts at $\alpha = x, y$ which probe the longitudinal or transversal excitations, respectively. Since we consider an isotropic Hamiltonian the DSF is invariant under the exchange of the $x$ and $y$ direction. In this case, it is expedient to define the transversal DSF as follows:

$$S^{xx+yy}(\omega, \mathbf{Q}) := S^{xx}(\omega, \mathbf{Q}) + S^{yy}(\omega, \mathbf{Q}) = -\frac{1}{\pi} \text{Im} \langle 0| S^-(-\mathbf{Q}) \frac{1}{\omega - (\mathscr{H}_{\text{eff}} - \mathscr{E}_0)} S^+(\mathbf{Q})|0\rangle , \tag{13}$$

combining the $x$ and $y$ contribution, see also Ref. [35]. From Eqs. (12) and (13) we learn that we have to compute the corresponding resolvents to determine their spectral densities.

Due to the conservation of the number of magnons in $\mathscr{H}_{\text{eff}}$ the Hilbert subspaces of different numbers of magnons do not interact. Hence, in subsequent calculations each subsector of $n$ magnons can be treated separately [18, 36]. This facilitates the computations greatly because it converts a true many-body problem into a few-body problem. For instance, the resolvent $R(\omega, \mathbf{Q})$ yielding the spectral density $S(\omega, \mathbf{Q}) = -\frac{1}{\pi} \text{Im} R(\omega, \mathbf{Q})$ splits into contributions from separate subspaces of different numbers $n$ of magnons

$$R(\omega, \mathbf{Q}) = \sum_n \langle 0| O^{(-n)}(-\mathbf{Q}) \frac{1}{\omega - (\mathscr{H}_{\text{eff}} - \mathscr{E}_0)} O^{(+n)}(\mathbf{Q})|0\rangle , \tag{14}$$

where $O^{(n)}$ stands for the part of the effective observable creating $n$ magnons (subscript $(+n)$) or annihilating $n$ magnons (subscript $(-n)$).

Since the original Hamilton operator changes the number of magnons only by pairs the effective observables which have contributions in the one-magnon sector, will generically also have contributions in the three-magnon sector and higher. The effective observables which have contributions in the two-magnon sector, will generically also have contributions in the four-magnon sector and higher. The transversal DSF $S^{xx+yy}(\omega, \mathbf{Q})$ results from $O = S_{\text{eff}}^\pm(\mathbf{Q})$

and couples to the odd sectors $n = 1$ and $n = 3$. Higher contributions, e.g., $n = 5$ are negligible as is supported by sum rules, see below. In the longitudinal DSF $S^{zz}(\omega, \mathbf{Q})$ the even sector $n = 2$ dominates which is again supported by a sum rule, see below. The associated *static* structure factors (SSF) are given by

$$S_n^{\alpha\alpha}(\mathbf{Q}) := \langle 0|S^\alpha(-\mathbf{Q})\mathscr{P}_n S^\alpha(\mathbf{Q})|0\rangle \ , \tag{15}$$

where $\mathscr{P}_n$ projects onto the subspace with $n$ magnons. Obviously, the SSFs provide the frequency integrals of the spectral densities of $R(\omega, \mathbf{Q})$ in (14).

If the resolvent in a subspace with more than a single magnon is computed, the interaction of each pair of magnons must be taken into account. Previously, this effect was not accounted for in spin wave calculations [35,37]. The description in terms of an effective $O(3)$-continuum model with adjusted parameters includes interaction effects, but is tailored to the Raman response, i.e., the response at zero momentum or infinite wavelength [38]. Hence the results computed for the microscopic lattice model by CST and subsequent treatment of the remaining few-body problem promote our understanding of the QASQ to a significantly higher level which has been beyond reach so far.

In the evaluation of the DSFs the dispersion and the effective magnon-magnon interaction have to be considered. Since we started from the Dyson-Maleev representation of the spin Hamiltonian (1) the Hamiltonian was not manifestly hermitian. This remains true during the CST and also after the transformation. The one-magnon part $\mathscr{H}_\mathrm{M}$ is hermitian, but the interaction part $\mathscr{V}^0$ is not. Thus no standard Lanczos algorithm is applicable, but a non-symmetric Lanczos algorithm [39] is applied to determine the continued fraction representation of the resolvent.

The magnon operators are represented on a mesh of discrete points in the MBZ. For the CST, this mesh cannot be chosen very dense because the number of differential equations to be solved becomes too large. In order to keep finite-size effects small in the calculation of spectral densities, we interpolate the coefficients in the Hamiltonian and the observables to obtain a denser mesh in momentum space. For the longitudinal channel, the system size is enhanced in this way from $L = 8$ to $L = 192$ where $L^2$ defines the number of points in the MBZ. For the continua in the transversal channel, we extrapolated from $L = 8$ to $L = 16$ which is sufficient in view of the dimension of the Hilbert space which is $L^4$ in the three-magnon channel because of the two undetermined momenta. Still, the discretization needs to be fine enough to capture the continua.

The resulting spectral densities are sums of weighted $\delta$-functions because we truncate the depth of the continued fractions where finite-size effects set it. Smooth densities are obtained by broadening the truncated continued fractions by replacing the $\delta$-peaks by Gaussians with the corresponding weight $W_i$ and a constant broadening $\sigma$

$$I(\omega) = \sum_i W_i \delta(\omega - \omega_i) \rightarrow \sum_i W_i \frac{1}{\sqrt{2\sigma^2\pi}} e^{\frac{1}{2}\left(\frac{\omega - \omega_i}{\sigma}\right)^2} \ . \tag{16}$$

There are three reasons for introducing this broadening. The first one is to account for the finite experimental resolution. The second one is to account for the experimental uncertainty in the determination of the continua. The measurement of continua is more challenging than the determination of pronounced peaks because the continua are more strongly affected by the possible systematic errors in the subtraction of the backgrounds. The third reason is to mimic the finite life time of the measured excitations induced by finite temperature and/or by imperfections (disorder) in the sample. Note that the latter two reasons suggest a broadening depending on the polarization because different polarizations focus on different physical processes, e.g., pronounced peaks are only discernible in the transverse response. Thus the transversal and the longitudinal response will generically show a different broadening.

The spectral weights depend slightly on the interpolation of the coefficients resulting from the numerical CST. In particular, the relative weights between channels of different magnon number are influenced since different interpolation schemes are employed for sectors of different magnon number. To ensure the correct relative weights, we rescale the resulting spectral densities such that the ratios between the weights, i.e., the frequency integrals, are consistent with the extrapolations of the static structure factors (15) which can be extrapolated reliably to the thermodynamic limit, i.e., to infinite system size.

## 3  Continuous transformation in momentum space

The continuous change of basis is parametrized by a running parameter $\ell$ starting at 0 and terminating the transformation at $\ell = \infty$. The reciprocal value of $\ell$ represents an energy cutoff. The flowing Hamiltonian $H(\ell)$ is transformed from the initial Hamiltonian $\mathscr{H}(\ell = 0) =: H_{\text{init}}$ to the final effective one $H(\ell \to \infty) =: \mathscr{H}_{\text{eff}}$. This is achieved by integrating the flow equation

$$\partial_\ell \mathscr{H}(\ell) = [\eta(\ell), \mathscr{H}(\ell)] , \tag{17}$$

using the particle conserving generator $\eta(\ell) = \mathscr{V}^+(\ell) - \mathscr{V}^-(\ell)$ [18, 40, 41]. This generator ensures that the effective Hamiltonian exhibits the desired form (9). All terms which net create or annihilate particles are rotated away in the course of the flow (17) [36, 40]. The resulting differential equations are presented in the Appendix A.2.3.

The relevant observables $O$, i.e., the spin operators $S^-(-\mathbf{Q})$, $S^+(-\mathbf{Q})$ and $S^z(\mathbf{Q})$, are transformed as well. In the framework of the CST this is achieved by applying the same generator

$$\partial_\ell O = [\eta(\ell), O(\ell)] . \tag{18}$$

as for transforming the Hamiltonian. The ensuing differential equations are given in the Appendix A.2.5. Eventually, we are in the position to determine the DSF at zero temperature and to compare it quantitatively with the measured counting rates of inelastic neutron scattering.

### 3.1  Truncation according to scaling dimension

In general, the commutator in the flow equation (17) generates operator terms that are not present in the initial Hamiltonian. For interacting many-body systems the exact treatment leads generically to an infinite number of operator terms in the Hamiltonian and in the observables during the flow. This is not tractable in practice.

In order to obtain a closed set of differential equations, it is necessary to truncate in some way. For a physically justified and systematically controlled truncation one has to classify the relevance of the different operator terms. A standard approach is to use a small expansion parameter, see for instance Refs. [40, 42]. For gapless systems such as long-range ordered quantum magnets whose elementary excitations are Goldstone bosons such a small parameter is not obvious. Common choices are to expand in powers of $1/S$ or of $1/N$ (here $N$ is the number of flavors) [22, 43]. We tried the $1/S$ expansion, but found it inefficient. On the one hand, it is necessary to include complicated hexatic terms, i.e., terms with six magnon operators, in order to be consistent. On the other hand, one does not capture the important renormalization of the magnon-magnon interactions, see below. Hence, we opt for the scaling dimension instead as truncation criterion [18].

Originally, the concept of scaling dimension was introduced to describe critical phenomena by means of renormalization group approaches and conformal field theories [44, 45]. It is designed to focus on the low-energy physics of a model. In the context of continuous transformations, it was previously used to treat the one-dimensional sine-Gordon model. The terms

of the operator product expansion of the vertex functions were classified according to their scaling dimension [46,47].

Yet one may wonder why a concept designed to describe the low-energy physics is appropriate for describing magnons at short wavelengths, i.e., at high energies? The argument runs as follows. For gapless magnons the energy threshold above which the multi-particle continua start, i.e., the lower band edge, is given by the single magnon dispersion. This is easy to see if the gapless mode occurs at zero momentum in the MBZ because the lower band edge of two-magnon states at momentum $\mathbf{q}$ is given by $\omega_2(\mathbf{q}) = \min_{\mathbf{k}}[\omega(\mathbf{q}-\mathbf{k}) + \omega(\mathbf{k})]$ which cannot be higher than $\omega(\mathbf{q})$. In fact, it equals $\omega(\mathbf{q})$ because otherwise the single magnon would decay into two-magnon states [41,48,49] for which there is no evidence [50]. Due to the bipartiteness of the square lattice, the single magnon cannot decay into two magnons, but may decay in at least three magnons, see the hybridization terms in Eq. (8). But this modifies the above argument only slightly such that the lower band edges of the three-magnon continuum reads

$$\omega_3(\mathbf{q}) = \min_{\mathbf{k}_1, \mathbf{k}_2}[\omega(\mathbf{q}-\mathbf{k}_1-\mathbf{k}_2) + \omega(\mathbf{k}_1) + \omega(\mathbf{k}_2)] \ . \tag{19}$$

Again, the continuum states close in energy to the dispersion are those where $\mathbf{k}_1$ and $\mathbf{k}_2$ are close to zero where the magnons are gapless. Hence, the hybridization of one-magnon and three-magnon states is strongly influenced by the physics at and close to zero wavevector. Thus, the scaling dimension is indeed an appropriate criterion for the relevance of certain physical processes.

If interaction and hybridization are taken into account, the multi-magnon states will be renormalized predominantly by the long wavelength excitations. We have to find the terms which remain important, if one zooms to smaller and smaller energies or momenta, respectively. This is expressed by the scaling dimension. We classify the relevance of operator terms by their scaling properties under the momentum transformation $\mathbf{k}_i \to \lambda \mathbf{k}_i$ with $\lambda < 1$ in the thermodynamic limit. We consider a generic term in $D$-dimensional momentum space

$$\mathcal{T} = \int_{\mathrm{BZ}} C_{\mathbf{k}_1\ldots\mathbf{k}_n} \mathcal{O}^n_{\mathbf{k}_1\ldots\mathbf{k}_n} \delta(\mathbf{k}_1 + \ldots + \mathbf{k}_n) d^D\mathbf{k}_1 \ldots d^D\mathbf{k}_n \ , \tag{20}$$

where $\mathcal{O}^n_{\mathbf{k}_1,\ldots\mathbf{k}_n}$ is a monomial of $n$ bosonic operators of creation or annihilation type. Conservation of momentum is ensured by the $\delta$-function $\delta(\mathbf{k}_1 + \ldots + \mathbf{k}_n)$.

If the momenta are rescaled one obtains the factor $\lambda^{D(\frac{n}{2}-1)}$ in front of $\mathcal{T}$ which is the dimensional contribution of the operator term. Moreover, one has to take into account the scaling properties of the coefficient $C_{\lambda\mathbf{k}_1\ldots\lambda\mathbf{k}_n} \to \lambda^c C_{\mathbf{k}_1\ldots\mathbf{k}_n}$ at small momenta defined by its leading behavior in the vicinity of $\mathbf{k}_i = \mathbf{0}$. Consequently, the operator term $\mathcal{T}$ acquires a total pre-factor $\lambda^{D(\frac{n}{2}-1)+c}$ where $D$ is the dimension, i.e., for the square lattice $D = 2$. Thus, the scaling dimension $d$ of $\mathcal{T}$ is defined by the exponent $d = (n-2) + c$. Obviously, operator terms with a larger scaling dimension become less important upon scaling with $\lambda < 1$. The crucial corollary is, that the relevance of $\mathcal{T}$ decreases with increasing number of its creation and annihilation operators. Hence, interactions and hybridizations between subspaces with higher and higher magnon number are less and less relevant.

For instance, the dispersion is given by $\omega = v_S |\mathbf{k}|$ for $|\mathbf{k}| \ll 1$ where $v_S$ is the spin wave velocity, i.e., $c = 1$. Hence, the single-magnon terms have scaling dimension $d = 1$. The quartic operator terms have scaling dimension $d = 2$ because the vertex functions [34] are bounded, i.e., $c = 0$. Hexatic terms have even higher scaling dimension 3 or higher, depending on $c$. This is the reason why we neglect them altogether in the present treatment.

In this article, we do not only track the flow of the Hamilton operator, but also the flow of the spin observables, which represent the vertex corrections. Thus the flow of the observables has to be truncated as well. The corresponding spin operators are not evaluated directly,

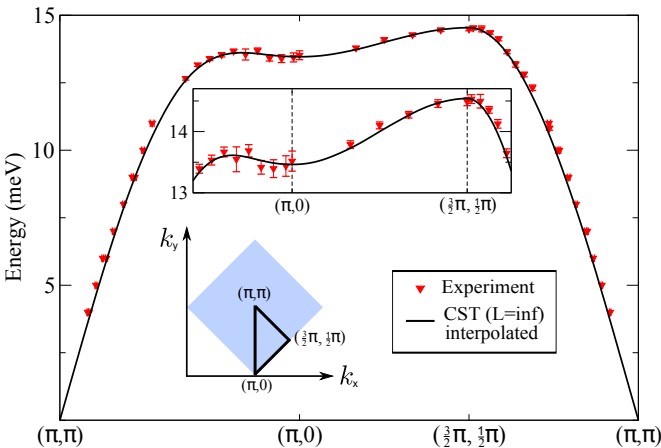

Figure 2: **One-magnon dispersion.** Dispersion along a representative path through the magnetic Brillouin zone (in light blue) comparing the dispersion obtained theoretically by CST with the experimental data (red triangles with error bars) from Ref. [16] for the coupling value $J = 6.11(2)$ meV. The inset shows the convincing agreement around the roton minimum at $(\pi, 0)$. The theoretical data are extrapolated to infinite system size on a mesh of wave vectors and interpolated between them which yields the solid line. (The lattice constant is set to unity.)

but as a part of a resolvent at zero temperature. It has turned out that it is an appropriate choice to use the number of excited (or de-excited) magnons as criterion for the truncation. This is in-line with our approach for the Hamiltonian where the scaling dimension essentially limits the number of bosonic operators in each monomial. For the transversal channel, in which the number of magnons is odd, we compute the one- and three-magnon channel. In the longitudinal channel where the number of magnons has to be even we study the two-magnon channel. This choice is strongly supported by the sum rules which agree very well with their theoretical values, see below. If we had omitted an important contribution the sum rules would indicate missing weight.

## 4 Spectroscopic signatures of interacting spin waves

Here we perform a detailed comparison between the theoretical results and the experimental data from inelastic scattering of polarized neutrons [15, 16]. First, we consider the applicability of the simple Heisenberg Hamiltonian (1). It is established that in the parent compounds of high-temperature superconductors further next-leading terms matter, see Ref. [33] and references therein, for instance ring exchange terms. But these terms are smaller by a factor of $t^2/U^2$ if $t$ is the hopping element and $U$ the on-site repulsion in an underlying Hubbard model. Since in the organic cuprate studied in Refs. [15, 16] the leading exchange $J$ is smaller than the exchange in the high-temperature cuprates by a factor 15 to 20, the hopping element $t$ must be smaller by about a factor 4. Hence the relative significance of such higher terms is suppressed by a factor 15 to 20 so that it ranges only on the percent level. For this reason, we do not consider it here, bearing in mind that quantitative conclusions may be influenced on the percent level.

Second, we determine the concrete value of the exchange coupling $J$, which defines the global energy scale. This can be done by fitting the energies at $\mathbf{k} = (\pi, 0)$ and $\mathbf{k} = (\frac{\pi}{2}, \frac{\pi}{2})$

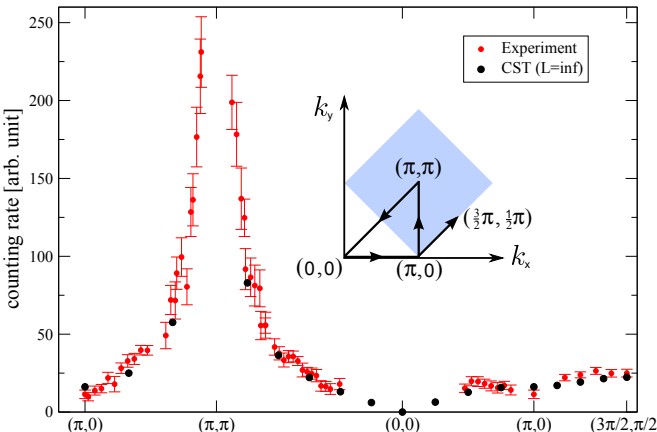

Figure 3: **Transversal spectral weight of the single magnon excitations.** Comparison of the measured weights from Ref. [15] with the computed weights $S_1^{xx}(\mathbf{Q}) + S_1^{yy}(\mathbf{Q})$, i.e., the transversal SSF, see Eq. (15). An overall proportionality factor is fitted.

yielding the value $J = 6.11(2)$ meV. The obtained excellent overall agreement of the dispersion is illustrated along a path along the high-symmetry lines of the Brillouin zone in Fig. 2.

We argued above that it is crucial to transform the observables as well. This means that the vertex corrections need to be taken into account. Fig. 3 compares the data obtained in this way with experimental results in the transversal channel. The agreement is very good and corroborates that the renormalized spin wave theory including the transformation of observables captures the relevant physics. This is supported further by the sum rules [51]. We checked the sum rules for $S = 1/2$ in the transversal channel

$$S_{\mathrm{t}}^{\mathrm{tot}} = \frac{1}{2} \int_{\mathrm{BZ}} d\mathbf{Q}(S^{xx}(\mathbf{Q}) + S^{yy}(\mathbf{Q})) = S \approx 0.495 \,, \tag{21}$$

and in the longitudinal channel

$$S_{\mathrm{l}}^{\mathrm{tot}} = \int_{BZ} d\mathbf{Q} S_{\mathrm{eff}}^{zz}(\mathbf{Q}) = S^2 \approx 0.2403 \,. \tag{22}$$

Note that the above integrals run over the full Brillouin zone, not the MBZ. The numerical values are obtained by extrapolating the effective observables to the thermodynamic limit. We emphasize that the nice agreement indicates that it is indeed quantitatively sufficient to consider the one-magnon and three-magnon contributions in the transversal channel and the two-magnon contribution in the longitudinal channel.

The main results for the DSFs are depicted in Fig. 4 where the dynamics in multi-magnon channels enters. To obtain these results we proceeded as follows, see also the Appendix B. The experimentally measured counting rates in the transversal channel at $\mathbf{k} = (\pi, 0)$ and $\mathbf{k} = (\frac{\pi}{2}, \frac{\pi}{2})$ (panels (c) and (d) in Fig. 4) exhibit pronounced peaks which we identify as the one-magnon peaks at the energies given by the dispersion $\omega_{\mathbf{k}}^{\mathrm{eff}}$ in (10).

Next, we address the broadening $\sigma$ on which the line shapes depend. We choose Gaussian broadening and convolve the theoretical results with the Gaussians to mimic various broadening mechanisms such as the finite experimental resolution, the uncertainties in the background subtraction, and finite residual temperature and disorder effects, see Eq. (16). Thus, the transversal broadening $\sigma_{\mathrm{t}}$ and the longitudinal broadening $\sigma_{\mathrm{l}}$ enter as additional fit parameters. The broadening and overall height of the line shapes is fitted to match the measured counting rates. The broadening is set to $\sigma_{\mathrm{t}} = 0.58(2)$ meV in the transversal channel and to

$\sigma_1 = 1.41(5)$ meV in the longitudinal channel. We recall, that the units of the energy axis and the counting rates are fixed for all displayed panels for both, the transversal and longitudinal channel.

The very good overall agreement between the theoretical curves and the experimental data in Figure 4 is convincing. First, we address the total DSF being the sum of the transversal and longitudinal DSFs. The positions and the heights of the pronounced one-magnon peaks as well as the continuum tails are captured by the theoretical line shapes in a quantitative way. The slight wiggles in the continua are due to the finite discretization in the Brillouin zone.

In the transversal channel one can discriminate the one-magnon contribution given by a Gaussian function (dashed magenta line) and the three-magnon continuum shaded in blue. An important experimental feature is the pronounced continuum tail at $\mathbf{k} = (\pi, 0)$. By contrast, the continuum a $\mathbf{k} = (\frac{\pi}{2}, \frac{\pi}{2})$ is much weaker; its spectral weight is marginal. Both aspects are captured with remarkable accuracy by the theoretical line shapes. The spectral weight in absolute units in the one-magnon channel is extrapolated for $L = \infty$ to be 0.5839 at $\mathbf{k} = (\frac{\pi}{2}, \frac{\pi}{2})$ and 0.4339 at $\mathbf{k} = (\pi, 0)$. The spectral weight in the three-magnon channel is extrapolated for $L = \infty$ to be 0.1337 and 0.2952, respectively. This implies that at $\mathbf{k} = (\frac{\pi}{2}, \frac{\pi}{2})$ 81.4% of the weight rest in the one-magnon peak while at $\mathbf{k} = (\pi, 0)$ it is only 59.5% due to the hybridization with the three-magnon continuum. Previously, the significant continuum at $\mathbf{k} = (\pi, 0)$ was interpreted as an indication of a fractionalization into spinons [16]. Further results on spectral weights in the various channels are given in the Appendix B.3.

To illustrate the crucial relevance of the magnon-magnon interaction, we determine the three-magnon continuum in the non-interacting case as well, i.e., we omit the interaction $\mathcal{V}^0$ in the effective model while leaving everything else unchanged. This yields the green curves. The difference between the magnon continuum for the interacting and non-interacting case at $\mathbf{k} = (\pi, 0)$ is striking. If the magnon-magnon interaction is omitted the spectral weight is shifted to significantly higher energies leading to a clear mismatch with the experimental findings.

The observations in the longitudinal channel are similar. The spectral weight in the two-magnon channel is found to be 0.2508 at $\mathbf{k} = (\frac{\pi}{2}, \frac{\pi}{2})$ and 0.2457 at $\mathbf{k} = (\pi, 0)$. The CST results agree very well with the experimental data. The very good accord is obviously spoiled by omitting the magnon-magnon interaction. Consequently, the pronounced signals in the measured intensities can be directly identified as the longitudinal magnon or Higgs resonance. The resonance is still fairly broad which implies that the longitudinal magnon lives only for a short time. The short life time can be traced back to the fact that the energy of the longitudinal magnon lies right within the two-magnon continuum into which it decays.

The important spectral weight found in the high-energy tails of the transversal spectral response results from the three-magnon continuum. This is a pivotal point because previous analyses take the continua as smoking gun of the relevance of a spinon scenario [14, 16]. Continuous contributions from three magnons only arise in the DSF from the vertex corrections, i.e., from the transformation of the spin observables. The hybridization terms have to be involved at least once in the CST of the observables. Otherwise, the Dyson-Maleev representation does not allow for a physical process linking two scattering states comprising three magnons each. This underlines the crucial progress achieved in the present work by the CST of the observables in comparison with the previous analysis [18].

The distribution of spectral weight *within* each continuum is a direct consequence of the attractive magnon-magnon interaction. The attractive magnon-magnon interaction shifts spectral weight *within* the three-magnon continuum to lower energies. By means of level repulsion, this in return decreases the energy of the single magnon states, i.e., the dispersion. This is the physical origin of the so-called roton dip in the dispersion at $\mathbf{k} = (\pi, 0)$. Qualitatively, this physics is also found by describing the Higgs resonance by so-called singlons [52]. We see that

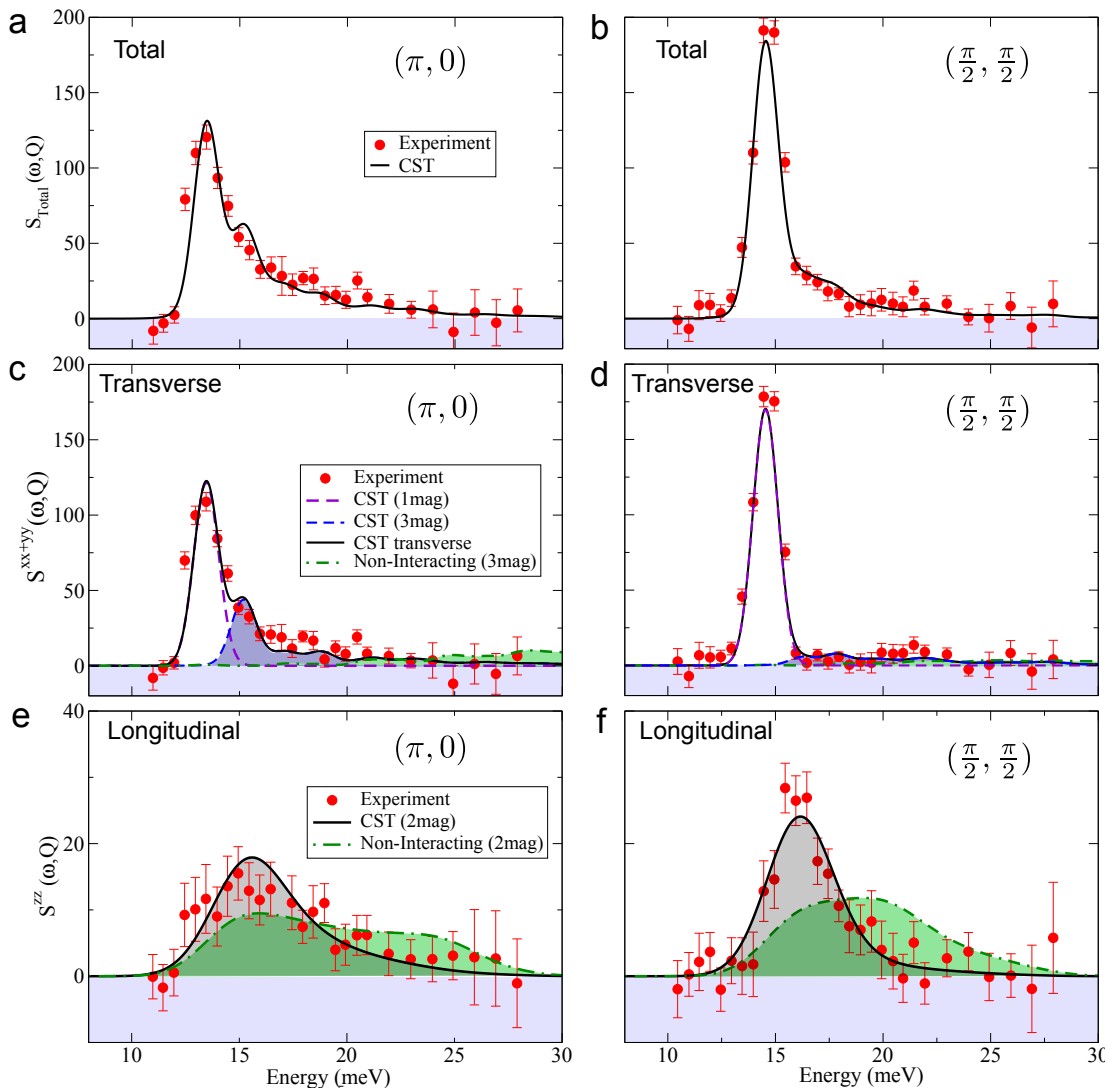

Figure 4: **Dynamic structure factors.** Comparison between the DSFs measured in Ref. [16] and the theoretical line shapes obtained from the CST. The energy scale is set to $J = 6.11(2)$meV for all curves. The arbitrary units on the $y$-axis are fixed globally to match the experimental data. The lattice constant is set to unity. (**a**) Total DSF (sum of transversal and longitudinal part) at $\mathbf{k} = (\pi, 0)$. (**b**) Total DSF at $\mathbf{k} = (\frac{\pi}{2}, \frac{\pi}{2})$. (**c**) Transversal DSF at $\mathbf{k} = (\pi, 0)$. The magenta line shows the dominant one-magnon peaks. The transversal broadening is set to $\sigma_\mathrm{t} = 0.58(2)$meV. The three-magnon continuum is shown as blue curve. Omitting the interaction leads to the green curve which does not agree with experiment at all. (**d**) Transverse DSF at $\mathbf{k} = (\frac{\pi}{2}, \frac{\pi}{2})$; otherwise same as in panel (**c**). (**e**) Longitudinal DSF at $\mathbf{k} = (\pi, 0)$. The longitudinal broadening is set to $\sigma_\mathrm{l} = 1.41(5)$meV to match the experimental data. The black line depicts the two-magnon continuum; the four-magnon contribution is negligible. The green line results from omitting the magnon-magnon interaction. (**f**) Longitudinal DSF at $\mathbf{k} = (\frac{\pi}{2}, \frac{\pi}{2})$; otherwise same as in panel (**e**).

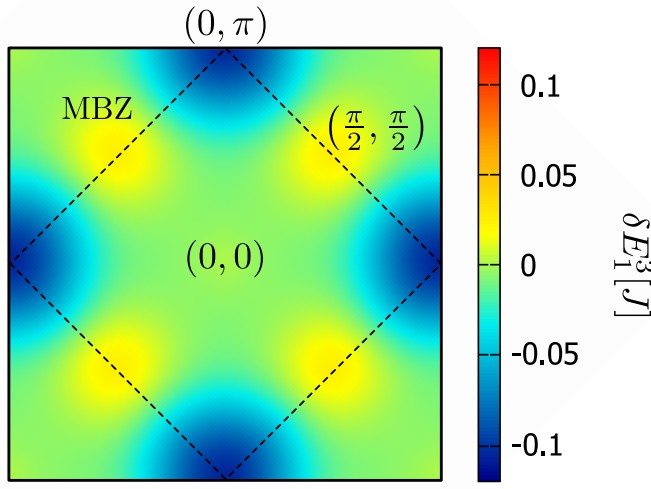

Figure 5: Shift of the dispersion given by (23) due to the flowing, i.e., renormalized, hybridization terms $\mathcal{V}^{\pm}$. Note that the shift is positive at wavevectors $\mathbf{k} = (\pm\frac{\pi}{2}, \pm\frac{\pi}{2})$ while it is clearly negative at $\mathbf{k} = (\pm\pi, 0)$ and $\mathbf{k} = (0, \pm\pi)$.

the size of the renormalized magnon-magnon interaction is very important. Thus, its quantitative renormalization matters and we pointed out previously, that it is enhanced by about 50% in the renormalization flow, see Supplement of Ref. [18].

To back the above given physical scenario we present a quantitative analysis of the interplay between the magnon-magnon attraction and the hybridization effects in the next section.

## 5 Non-perturbative renormalization of high-energy magnons

Each type of operator term stands for a particular physical process. The solution of the flow equations, given in the Appendix A, tells us how the prefactor of such a physical process and hence its significance changes in the course of the flow, i.e., how it is renormalized. We use this information to determine the relevance of different physical processes and how they influence the effective Hamiltonian (9) at the end of the CST. An important example is the hybridization between one- and three-magnon states. We find that the hybridization between one and three-magnon states contributes significantly to the renormalization of the anomalous high-energy dispersion.

The contribution of the hybridization to the one-magnon dispersion is given by the term

$$\delta E_1^3(\mathbf{k}) = -4 \int_0^\infty d\ell \sum_{1,2,3} \delta_{\mathbf{k}-3}^{12} \mathcal{V}_{\mathbf{k}-321}^{(\alpha-3)}(\ell) \mathcal{V}_{12-3\mathbf{k}}^{(3-\alpha)}(\ell) \,, \tag{23}$$

which stems from the corresponding summand in the flow of the one-magnon dispersion $\partial_\ell \omega_{\mathbf{k}}(\ell)$. The resulting shift of the dispersion is depicted in Fig. 5. Clearly, this explains the occurrence of the roton dips because the dispersion is lifted upward at wavevectors $\mathbf{k} = (\pm\frac{\pi}{2}, \pm\frac{\pi}{2})$. In contrast, it is pushed downwards at wavevectors $\mathbf{k} = (\pm\pi, 0)$ and $\mathbf{k} = (0, \pm\pi)$.

To elucidate the significance and the origin of $\delta E_1^3(\mathbf{k})$ further we consider a perturbative evaluation of (23) using the conventional spin wave expansion in $1/S$ first. Subsequently we interpret the full non-perturbative evaluation of (23). In leading order in $1/S$ the flow of the

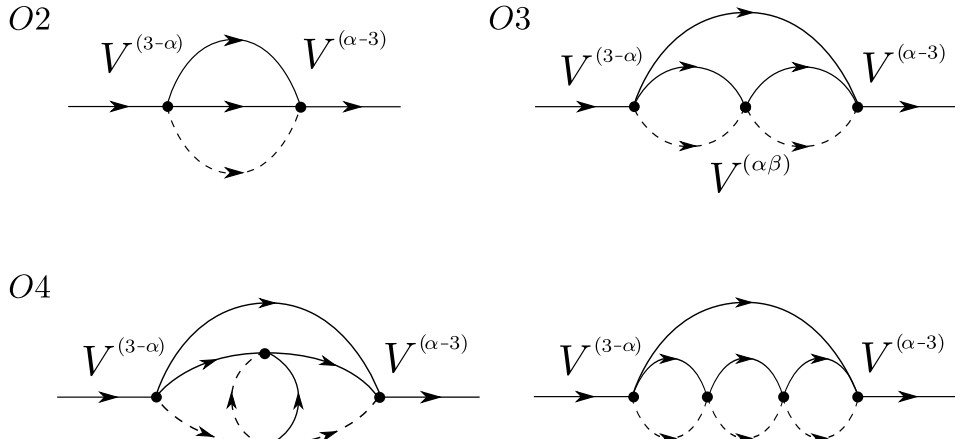

Figure 6: Three contributions $\delta E_1^3(\mathbf{k})$ from the hybridization terms in (8) to the renormalized one-magnon dispersion $\omega_{\mathbf{k}}^{\text{eff}}$ as given in (23). The solid lines stand for the propagation of an $\alpha$-magnon while the dashed lines stand for the propagation of a $\beta$-magnon. The upper left diagram is second order (O2) in the quartic terms $V$. The upper right diagram is third order (O3) in the quartic terms $V$; the additional order results from an interaction between an $\alpha$ and a $\beta$ magnon. The lower left diagram is fourth order (O4) in the quartic terms $V$; the two additional orders result from two hybridization vertices (8). The lower right diagram is also of fourth order; it represents an iterated interaction between the $\alpha$- and $\beta$-magnon.

coefficients is given by

$$\mathscr{V}_{\mathbf{k-321}}^{(\alpha-3)}(\ell) = V_{\mathbf{k-321}}^{(\alpha-3)} \exp\left[-(\omega_{\mathbf{k}}^{(0)} - \omega_1^{(0)} - \omega_2^{(0)} - \omega_3^{(0)})\ell\right], \tag{24a}$$

$$\mathscr{V}_{\mathbf{12-3k}}^{(3-\alpha)}(\ell) = V_{\mathbf{12-3k}}^{(3-\alpha)} \exp\left[-(\omega_{\mathbf{k}}^{(0)} - \omega_1^{(0)} - \omega_2^{(0)} - \omega_3^{(0)})\ell\right], \tag{24b}$$

so that the leading term of $\delta E_1^3(\mathbf{k})$ is second order in $V$. Using (24) we obtain

$$\delta E_1^3(\mathbf{k}) = \sum_{1,2,3} \frac{2V_{\mathbf{k-321}}^{(\alpha-3)} V_{\mathbf{12-3k}}^{(3-\alpha)} \delta_{\mathbf{k-3}}^{\mathbf{1+2}}}{\omega_{\mathbf{k}}^{(0)} - \omega_1^{(0)} - \omega_2^{(0)} - \omega_3^{(0)}}, \tag{25}$$

which equals the result of second order perturbation theory in $V$ specified by the upper left diagram O2 shown in Fig. 6.

In third order, the upper right diagram O3 in Fig. 6, the attractive interaction between an $\alpha$-magnon and a $\beta$-magnon is included. Such a diagram is taken into account in a full perturbative calculation up to $1/S^3$ as carried out by Syromyatnikov [31]. Further contributions lead to higher corrections such as depicted in the diagrams O4 in the second row of Fig. 6. Interestingly, the left O4 diagram results from the renormalization of the magnon-magnon interaction while the right O4 diagram results from the renormalization of the hybridization terms. Physically, it represents the repeated interaction processes between an $\alpha$-magnon and a $\beta$-magnon.

Our results, in combination with the slow convergence found in the direct perturbative approach [31], indicate that the O4 and higher terms are quantitatively significant. In our non-perturbative renormalizing approach the flow of all coefficients up to scaling dimension $d = 2$ is evaluated including all mutual dependencies. As a result, the term in (23) includes vertex corrections up to infinite order in $1/S$ as illustrated in Fig. 7. The interaction and

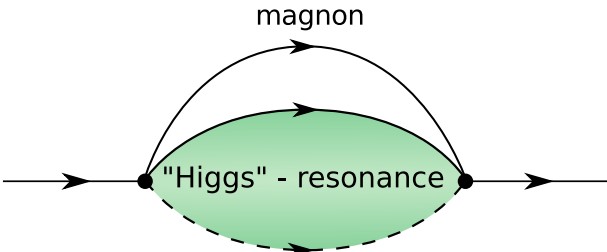

Figure 7: Sum of all contributions to $\delta E_1^3(\mathbf{k})$ from the renormalization of the propagation of a pair of $\alpha$- and $\beta$-magnon which eventually form the Higgs resonance.

hybridization processes renormalize the propagation of a pair of $\alpha$- and $\beta$-magnon which form the Higgs resonance [18]. We stress, however, that there are also processes linking the upper $\alpha$-magnon and the $\beta$-magnon and processes linking the two $\alpha$-magnons (both not included in the diagram in Fig. 7). The latter are of minor importance, see below. But all these processes are included in the solution of the flow equation (17).

The significance of the renormalization of the attractive interaction between an $\alpha$- and a $\beta$-magnon is strongly corroborated by the fact that the downshift of the dispersion at the roton minimum diminishes considerably (from 8% to 5%) if we switch off the flow of the interaction by artificially setting $\partial_\ell \mathscr{V}^{\alpha\beta}(\ell) = 0$. By contrast, the renormalization of the interaction between magnons of the same sublattice (given by $\mathscr{V}^{\alpha\alpha}$ or $\mathscr{V}^{\beta\beta}$) does not affect the high-energy dispersion significantly.

Finally, in order to quantify the energy reduction induced by the magnon-magnon interaction $\mathscr{V}^{\alpha\beta}$ we diagonalize the two-magnon channel with one $\alpha$-magnon and one $\beta$-magnon for finite system sizes. The resulting spectrum is compared with the non-interacting case $\mathscr{H}_{\text{SW}}^{\text{eff}}(\mathbf{k})$ where the energy spectrum is simply given by the sum of two one-magnon energies, i.e., two values of the dispersion, with given total momentum $\mathbf{k}$. Then, we define the difference between the next-lowest energy levels in these two cases given by

$$\Delta E^{2\text{-mag}}(\mathbf{k}) := E_1\left(\mathscr{H}_{\text{SW}}^{\text{eff}}(\mathbf{k})\right) - E_1\left(\mathscr{H}^{\text{eff}}(\mathbf{k})\right) , \tag{26}$$

where $E_i(\mathscr{H})$ denotes the $i$-th eigenvalue of $\mathscr{H}$ sorted in ascending order counting degenerate eigenenergies only once. Note that the lowest energy level $E_0(\mathbf{k})$ is defined by the one-magnon energy $\omega_{\mathbf{k}}^{\text{eff}}$ representing the lower band edge of the two-magnon continuum. This band edge is by construction unaffected by the interaction. Hence it does not provide any information on attractive forces or binding so that we resorted to $E_1$ to define $\Delta E^{2\text{-mag}}(\mathbf{k})$.

Fig. 8 indicates where spectral weight is shifted downwards in the Brillouin zone and to what extent. Keeping the scale in mind one realizes that there is a general trend of downshifting. But the downshifts are clearly maximum around $\mathbf{k} = (\pm\pi, 0)$ and $\mathbf{k} = (0, \pm\pi)$ and smaller around at $\mathbf{k} = (\pm\frac{\pi}{2}, \pm\frac{\pi}{2})$. Thus, a comparison of Figs. 5 and 8 shows very similar qualitative features. This observation underlines our interpretation that one-magnon states hybridize with three-magnon states which are essentially built from single magnons and a Higgs resonance (or longitudinal magnon). As the energy of the Higgs resonance is lowered at $\mathbf{k} = (\pi, 0)$ and its equivalent wavevectors the corresponding magnon-Higgs continuum is repelling the energy level of the single-magnon state, see Fig. 7. Our results indicate that it is mandatory to track the full renormalization of the interaction and the hybridization terms in order to capture the physics quantitatively.

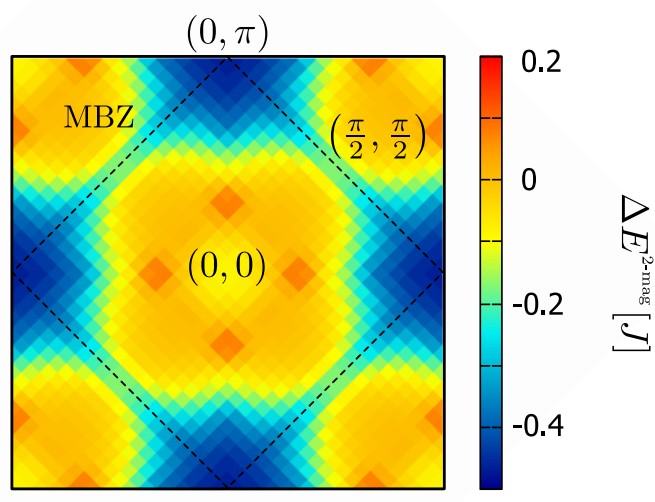

Figure 8: Energy difference as defined in Eq. (26) between the lowest energy level in the longitudinal channel, i.e., between one $\alpha$-magnon and one $\beta$-magnon, with and without interaction.

## 6 Conclusions

Summarizing, our detailed analysis shows that the effective magnon model obtained by renormalizing via a continuous basis change captures the physics of the long-range ordered Heisenberg model on a square lattice quantitatively. This is in stark contrast to a perturbative treatment expanding in $1/S$ which converges slowly. Including also the appropriate renormalization of the observables, i.e., including the relevant vertex corrections within the CST formalism, allows us to obtain an impressive agreement with experiment. This holds for both channels, transversal and longitudinal with respect to the staggered magnetization which is the order parameter.

In particular, the much debated continua can be reproduced. In order to retrieve the weight in the three-magnon continuum occurring in the transversal DSF, the proper treatment of the transformations of the observables is decisive. For the distribution of the weights in the two-magnon longitudinal and in the three-magnon transversal continua it is important to take the renormalized attractive magnon-magnon interaction properly into account. To capture the continua properly is a crucial aspect because previously the significant continua were interpreted as evidence for a failure of the magnon description, justifying to search beyond the Goldstone bosons for qualitatively different fractional excitations such as spinons. We stress that our approach yields a very good agreement with experimental data without resorting to spinons.

In conclusion, the Goldstone bosons of the long-range ordered two-dimensional Heisenberg model not only describe the dynamics at low energies, but also at high energies if the magnon-magnon interaction is taken into account quantitatively. Nevertheless, the magnon picture of the square-lattice Heisenberg model seems to be fragile to additional four-spin interactions as recent quantum Monte Carlo simulations suggest [53]. Thus, the authors of Ref. [53] regard the high-energy magnons close to the roton minimum as pairs of nearly deconfined spinons in the pure square-lattice Heisenberg model. We stress that this characterization is not in contradiction to our findings. It would be very interesting to study the influence of such four-spin interactions on the magnons with our CST formalism.

With respect to high $T_c$ superconductivity, the pure magnetic side of the long-standing

quest for understanding the underlying physics appears to be solved. This provides a firm basis to tackle the ensuing hole-magnon interaction and the induced hole-hole interaction in future studies.

In view of the wide-spread presence of long-range magnetic order in general, the continuous similarity transformation introduced here in great detail provides a powerful tool to study the dynamics of the elementary excitations, the Goldstone magnons, in such systems. This includes the important effective magnon-magnon interaction as well as the vertex corrections describing the effective observables. Consequently, we expect future applications to a variety of fascinating physical systems such as two- and three-dimensional quantum magnets close to quantum criticality where the Higgs amplitude plays an even more important role in the dynamical correlation functions [54–58].

# Acknowledgements

This work was supported by the Cusanuswerk (MP) and by the Deutsche Forschungsgemeinschaft and the Russian Foundation of Basic Research in TRR 160. We thank N. Christensen, B. Normand, H. Rønnow, A. Sandvik, R. Singh, and A. Syromyatnikov for fruitful discussions and exchange of data.

# A Derivation of the effective magnon description

In the following part, we explicate technical details concerning the derivation of the effective magnon description for the Heisenberg quantum antiferromagnet with $S = 1/2$ on the square lattice using a particle conserving continuous similarity transformation (CST).

## A.1 Dyson-Maleev representation

The transformation itself has been given in the main text in Eq. (2) and the resulting Hamilton operator in Eqs. (4) to (8).

### A.1.1 Observables

The operators which are required for the evaluation of the dynamic structure factors are obtained by the transformation in Eq. (2) of the main text. For the longitudinal part, we consider the operator

$$
\begin{aligned}
S^z(\mathbf{Q}) = S_A^z(\mathbf{Q}) + S_B^z(\mathbf{Q}) = &\left( m_{\mathbf{Q}}^2 - SN \right)\left( \Gamma_{\mathbf{Q}} - 1 \right) \delta_G(\mathbf{Q}) \\
&+ \sum_{1,2} \delta_G(\mathbf{Q} - 1 + 2) \left\{ [\Gamma_{\mathbf{K}} l_1 l_2 - m_1 m_2] \beta_1^\dagger \beta_2 + [\Gamma_{\mathbf{K}} m_1 m_2 - l_1 l_2] \alpha_1^\dagger \alpha_2 \right. \\
&\left. + [\Gamma_{\mathbf{K}} m_1 l_2 - l_1 m_2] \alpha_1^\dagger \beta_{-2}^\dagger + [\Gamma_{\mathbf{K}} m_2 l_1 - l_2 m_1] \alpha_2 b_{-1} \right\}, \quad (27)
\end{aligned}
$$

with the factor $\Gamma_{\mathbf{K}} = \gamma(\mathbf{Q} - 1 + 2)$ which tracks the reciprocal lattice vector $\mathbf{K} = \mathbf{Q} - 1 + 2$. Depending on the MBZ in which this vector $\mathbf{K} \in \Gamma_A^*$ is located the function takes the values $\Gamma_{\mathbf{K}} = 1$ or $\Gamma_{\mathbf{K}} = -1$. It is positive in the first MBZ and negative in the adjacent edge-sharing MBZs, i.e., it switches sign each time one enters another MBZ across an edge as shown in Fig. 9.

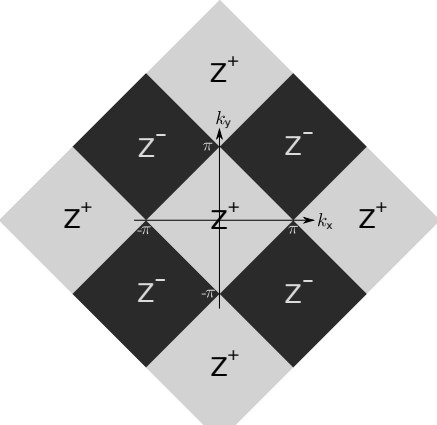

Figure 9: Illustration of the sign structure of the factor $\Gamma_{\mathbf{K}}$. The factor takes the value $\Gamma_{\mathbf{K}} = +1$ for $\mathbf{K} \in \mathbf{Z}^+$ or $\Gamma_{\mathbf{K}} = +1$ for $\mathbf{K} \in \mathbf{Z}^-$.

For the transversal part we transform the operators

$$
S^+(\mathbf{Q}) = \sqrt{2SN}\left(1 - \frac{1}{SN}\sum_{\mathbf{1}} m_{\mathbf{1}}^2\right)\left(\left(l_{\mathbf{Q}} + \Gamma_{\mathbf{Q}} m_{\mathbf{Q}}\right)\alpha_{-\mathbf{Q}} + \left(m_{\mathbf{Q}} + \Gamma_{\mathbf{Q}} l_{\mathbf{Q}}\right)\beta_{\mathbf{Q}}^\dagger\right)
$$
$$
-\frac{1}{\sqrt{2SN}}\sum_{\mathbf{1,2,3}}\delta_G(\mathbf{Q} - \mathbf{1} + \mathbf{2} + \mathbf{3})\Big\{\left[l_{\mathbf{1}}l_{\mathbf{2}}l_{\mathbf{3}} + \Gamma_{\mathbf{K}} m_{\mathbf{1}}m_{\mathbf{2}}m_{\mathbf{3}}\right]\alpha_{\mathbf{1}}^\dagger\alpha_{\mathbf{2}}\alpha_{\mathbf{3}}
$$
$$
+\left[m_{\mathbf{1}}l_{\mathbf{2}}l_{\mathbf{3}} + \Gamma_{\mathbf{K}} m_{\mathbf{2}}m_{\mathbf{3}}l_{\mathbf{1}}\right]\alpha_{\mathbf{2}}\alpha_{\mathbf{3}}\beta_{-\mathbf{1}} + 2\left[l_{\mathbf{1}}l_{\mathbf{2}}m_{\mathbf{3}} + \Gamma_{\mathbf{K}} m_{\mathbf{2}}l_{\mathbf{3}}m_{\mathbf{1}}\right]\alpha_{\mathbf{1}}^\dagger\alpha_{\mathbf{2}}\beta_{-\mathbf{3}}^\dagger
$$
$$
+2\left[m_{\mathbf{1}}l_{\mathbf{2}}m_{\mathbf{3}} + \Gamma_{\mathbf{K}} l_{\mathbf{1}}m_{\mathbf{2}}l_{\mathbf{3}}\right]\alpha_{\mathbf{2}}\beta_{-\mathbf{3}}^\dagger\beta_{-\mathbf{1}} + \left[l_{\mathbf{1}}m_{\mathbf{2}}m_{\mathbf{3}} + \Gamma_{\mathbf{K}} m_{\mathbf{1}}l_{\mathbf{2}}l_{\mathbf{3}}\right]\alpha_{\mathbf{1}}^\dagger\beta_{-\mathbf{2}}^\dagger\beta_{-\mathbf{3}}^\dagger
$$
$$
+\left[m_{\mathbf{1}}m_{\mathbf{2}}m_{\mathbf{3}} + \Gamma_{\mathbf{K}}l_{\mathbf{1}}l_{\mathbf{2}}l_{\mathbf{3}}\right]\beta_{-\mathbf{2}}^\dagger\beta_{-\mathbf{3}}^\dagger\beta_{-\mathbf{1}}\Big\}\,, \quad (28)
$$

and

$$
S^-(\mathbf{Q}) = \sqrt{2SN}\left(\left[l_{\mathbf{Q}} + \Gamma_{\mathbf{Q}}m_{\mathbf{Q}}\right]\alpha_{\mathbf{Q}}^\dagger + \left[m_{\mathbf{Q}} + \Gamma_{\mathbf{Q}}l_{\mathbf{Q}}\right]\beta_{-\mathbf{Q}}\right)\,. \quad (29)
$$

Note, that the external momentum $\mathbf{Q}$ can also take values outside the first MBZ and that $S(\mathbf{Q}) \neq S(\mathbf{Q} + \mathbf{g})$. Due to the non-hermiticity of the Dyson Maleev representation we have $S(-\mathbf{Q})^- \neq \left(S(\mathbf{Q})^+\right)^\dagger$ and, thus, both operators have to be transformed independently.

## A.2 Continuous similarity transformation: CST

### A.2.1 General approach

The particular form of the flowing Hamiltonian and observables is defined in second quantization as an expansion in terms of normal-ordered operators. We write

$$
H(\ell) = \sum_i h_i(\ell)A_i\,, \quad O(\ell) = \sum_i o_i(\ell)D_i\,, \quad (30)
$$

with constant monomials of bosonic operators $A_i, D_i$ and $\ell$-dependent coefficients $h_i(\ell), o_i(\ell)$. In this way, the flow equations turn into a system of differential equations for the scalar coefficients $h_i(\ell)$ and $o_i(\ell)$. In order to determine these equations, we evaluate the commutators $[\eta(\ell), H(\ell)]$ and $[\eta(\ell), O(\ell)]$, normal-order the results and equate the coefficients on both sides of the flow equation. In general, new operators $A_i$ and $D_i$ will occur in the commutators, which are not present in the initial Hamiltonian and observable. The proliferation of such

operators leads to an infinitely large operator basis. In order to obtain a closed set of differential equations, a systematic and physically justified truncation scheme is required. Here, we use the scaling dimension of operator terms as a truncation criterion because it is particularly suitable to gapless systems, see main text and Ref. [18].

### A.2.2  Truncation of the Hamiltonian: Scaling dimension

Here, we take into account the flow of operator terms up to a scaling dimension of $d = 2$. Thus, we may neglect hexatic operator terms with $n = 6$ which have scaling dimension $d = 4$. The resulting flowing Hamiltonian reads

$$
\begin{aligned}
H(\ell) = {}& E_0 + \sum_{1} \omega(\mathbf{1}) \left( a_1^\dagger a_1 + b_1^\dagger b_1 \right) + \Gamma(\mathbf{1}) \left( a_1^\dagger b_{-1}^\dagger + a_1 b_{-1} \right) \\
& + \sum_{1,2,3,4} \Big\{ C_1(\mathbf{1,2,3,4}) a_1^\dagger a_2^\dagger a_3 a_4 + C_2(\mathbf{1,2,3,4}) a_1^\dagger a_2 b_3^\dagger b_4 + C_3(\mathbf{1,2,3,4}) b_1^\dagger b_2^\dagger b_3 b_4 \\
& \quad + C_4(\mathbf{1,2,3,4}) a_1^\dagger a_2^\dagger a_3 b_4^\dagger + C_5(\mathbf{1,2,3,4}) a_1^\dagger b_2^\dagger b_3^\dagger b_4 + C_6(\mathbf{1,2,3,4}) a_1^\dagger a_2 a_3 b_4 \\
& \quad + C_7(\mathbf{1,2,3,4}) a_1 b_2^\dagger b_3 b_4 + C_8(\mathbf{1,2,3,4}) a_1^\dagger a_2^\dagger b_3^\dagger b_4^\dagger + C_9(\mathbf{1,2,3,4}) a_1 a_2 b_3 b_4 \Big\}. \quad (31)
\end{aligned}
$$

Then, the flowing generator $\eta(\ell)$ is given by

$$
\begin{aligned}
\eta(\ell) = {}& \sum_{1} \Gamma(\mathbf{1}) \left( a_1^\dagger b_{-1}^\dagger - a_1 b_{-1} \right) \\
& + \sum_{1,2,3,4} \Big\{ C_4(\mathbf{1,2,3,4}) a_1^\dagger a_2^\dagger a_3 b_4^\dagger + C_5(\mathbf{1,2,3,4}) a_1^\dagger b_2^\dagger b_3^\dagger b_4 - C_6(\mathbf{1,2,3,4}) a_1^\dagger a_2 a_3 b_4 \\
& \quad - C_7(\mathbf{1,2,3,4}) a_1 b_2^\dagger b_3 b_4 + C_8(\mathbf{1,2,3,4}) a_1^\dagger a_2^\dagger b_3^\dagger b_4^\dagger - C_9(\mathbf{1,2,3,4}) a_1 a_2 b_3 b_4 \Big\}. \quad (32)
\end{aligned}
$$

The coefficients $E_0$, $\omega(\mathbf{1})$, $\Gamma(\mathbf{1})$, and $C_i(\mathbf{1,2,3,4})$ depend on the flow parameter $\ell$ and satisfy the initial conditions

$$
E_0\big|_{\ell=0} = -2J(S^2 + AS + A^2/4)\,, \tag{33a}
$$

$$
\omega(\mathbf{1})\big|_{\ell=0} = 2J(2S + A)\sqrt{1 - \gamma_1^2}\,, \tag{33b}
$$

$$
\Gamma(\mathbf{1})\big|_{\ell=0} = 0\,, \tag{33c}
$$

$$
C_1(\mathbf{1,2,3,4})\big|_{\ell=0} = -l_1 l_2 l_3 l_4 \frac{J}{N} V^{(1)}_{1234} \delta_G(\mathbf{1+2-3-4})\,, \tag{33d}
$$

$$
C_2(\mathbf{1,2,3,4})\big|_{\ell=0} = -4 l_1 l_2 l_3 l_4 \frac{J}{N} V^{(4)}_{1\text{-}42\text{-}3} \delta_G(\mathbf{1-2+3-4})\,, \tag{33e}
$$

$$
C_3(\mathbf{1,2,3,4})\big|_{\ell=0} = -l_1 l_2 l_3 l_4 \frac{J}{N} V^{(9)}_{\text{-}4\text{-}3\text{-}2\text{-}1} \delta_G(\mathbf{1+2-3-4})\,, \tag{33f}
$$

$$
C_4(\mathbf{1,2,3,4})\big|_{\ell=0} = -2 l_1 l_2 l_3 l_4 \frac{J}{N} V^{(3)}_{12\text{-}43} \delta_G(\mathbf{1+2-3+4})\,, \tag{33g}
$$

$$
C_5(\mathbf{1,2,3,4})\big|_{\ell=0} = -2 l_1 l_2 l_3 l_4 \frac{J}{N} V^{(6)}_{\text{-}4\text{-}1\text{-}2\text{-}3} \delta_G(\mathbf{1+2+3-4})\,, \tag{33h}
$$

$$
C_6(\mathbf{1,2,3,4})\big|_{\ell=0} = -2 l_1 l_2 l_3 l_4 \frac{J}{N} V^{(2)}_{1\text{-}423} \delta_G(\mathbf{1-2-3-4})\,, \tag{33i}
$$

$$
C_7(\mathbf{1,2,3,4})\big|_{\ell=0} = -2 l_1 l_2 l_3 l_4 \frac{J}{N} V^{(5)}_{\text{-}4\text{-}31\text{-}2} \delta_G(\mathbf{-1+2-3-4})\,, \tag{33j}
$$

$$
C_8(\mathbf{1,2,3,4})\big|_{\ell=0} = -l_1 l_2 l_3 l_4 \frac{J}{N} V^{(7)}_{12\text{-}3\text{-}4} \delta_G(\mathbf{1+2+3+4})\,, \tag{33k}
$$

$$
C_9(\mathbf{1,2,3,4})\big|_{\ell=0} = -l_1 l_2 l_3 l_4 \frac{J}{N} V^{(8)}_{\text{-}3\text{-}412} \delta_G(\mathbf{-1-2-3-4})\,. \tag{33l}
$$

The explicit vertex functions $V^{(i)}_{1234}$ are defined in Ref. [34].

### A.2.3 Flow equations: Hamiltonian

Inserting $H(\ell)$ and $\eta(\ell)$ into the flow equation $\partial_\ell H(\ell) = [\eta(\ell), H(\ell)]$ and keeping all operators up to scaling dimension $d = 2$, we obtain the following differential equations

$$\partial_\ell E_0 = -8 \sum_{1,2,3,4} C_8(\mathbf{1,2,3,4}) C_9(\mathbf{1,2,3,4}) \delta_G(\mathbf{1+2+3+4}) + -2 \sum_1 \Gamma(\mathbf{1}) \Gamma(\mathbf{1}) , \quad (34a)$$

$$\partial_\ell \omega(\mathbf{1}) = (-2) \Gamma(\mathbf{1}) \Gamma(\mathbf{1}) + \sum_{3,4} \Big\{ (-4) C_4(\mathbf{3,4,1,5}) C_6(\mathbf{1,3,4,5}) \delta_G(\mathbf{3+4-1+5})$$

$$+ (-16) C_8(\mathbf{1,3,4,5}) C_9(\mathbf{1,3,4,5}) \delta_G(\mathbf{3+4+1+5}) \Big\}$$

$$+ \sum_3 \Big\{ (-4) C_8(\mathbf{1,3,1,-3}) \Gamma(\mathbf{3}) - 4 C_6(\mathbf{1,1,3,-3}) \Gamma(\mathbf{3}) \Big\} \quad (34b)$$

$$\partial_\ell \Gamma(\mathbf{1}) = (-2) \Gamma(\mathbf{1}) \omega(\mathbf{1}) + \sum_{3,4,5} \Big\{ (-8) C_4(\mathbf{3,4,1,5}) C_9(\mathbf{3,4,-1,5}) \delta_G(\mathbf{-1+3+4+5})$$

$$+ (-8) C_5(\mathbf{3,4,5,-1}) C_9(\mathbf{1,3,4,5}) \delta_G(\mathbf{1+3+4+5}) \Big\}$$

$$+ \sum_3 \Big\{ (-1) C_2(\mathbf{1,3,-1,-3}) \Gamma(\mathbf{3}) + (-8) C_8(\mathbf{1,3,-1,-3}) \Gamma(\mathbf{3}) \Big\} , \quad (34c)$$

$$\partial_\ell C_1(\mathbf{1,2,3,4}) = \delta_G(\mathbf{1+2-3-4}) \Big\{ (-1) C_4(\mathbf{1,2,4,-3}) \Gamma(\mathbf{3}) + (-1) C_4(\mathbf{1,2,3,-4}) \Gamma(\mathbf{4})$$

$$+ (-1) C_6(\mathbf{2,3,4,-1}) \Gamma(\mathbf{1}) + (-1) C_6(\mathbf{1,3,4,-2}) \Gamma(\mathbf{2}) +$$

$$\sum_{5,6} (-2) C_4(\mathbf{2,5,4,6}) C_6(\mathbf{1,3,5,6}) \delta_G(\mathbf{2+5-4+6}) +$$

$$(-2) C_4(\mathbf{1,5,3,6}) C_6(\mathbf{2,4,5,6}) \delta_G(\mathbf{1+5-3+6}) +$$

$$(-2) C_4(\mathbf{2,5,3,6}) C_6(\mathbf{1,4,5,6}) \delta_G(\mathbf{1+5-3+6}) +$$

$$(-2) C_4(\mathbf{1,5,4,6}) C_6(\mathbf{2,3,5,6}) \delta_G(\mathbf{1+5-3+6}) +$$

$$(-4) C_8(\mathbf{1,2,5,6}) C_9(\mathbf{3,4,5,6}) \delta_G(\mathbf{1+2+5+6}) \Big\} , \quad (34d)$$

$$\partial_\ell C_2(\mathbf{1,2,3,4}) = \delta_G(\mathbf{1-2+3-4}) \Big\{ (-4) C_4(\mathbf{1,-4,2,3}) \Gamma(\mathbf{-4}) + (-4) C_5(\mathbf{1,3,-2,4}) \Gamma(\mathbf{2})$$

$$+ (-4) C_6(\mathbf{1,2,-3,4}) \Gamma(\mathbf{-3}) + (-4) C_7(\mathbf{2,3,4,-1}) \Gamma(\mathbf{1}) +$$

$$\sum_{5,6} (-4) C_4(\mathbf{5,6,2,3}) C_6(\mathbf{1,5,6,4}) \delta_G(\mathbf{5+6-2+3})$$

$$+ (-4) C_5(\mathbf{1,5,6,4}) C_7(\mathbf{2,3,5,6}) \delta_G(\mathbf{1+5+6-4})$$

$$+ (-8) C_4(\mathbf{1,5,2,6}) C_7(\mathbf{5,3,4,6}) \delta_G(\mathbf{1+5-2+6})$$

$$+ (-8) C_5(\mathbf{5,3,6,4}) C_6(\mathbf{1,2,5,6}) \delta_G(\mathbf{5+3+6-4})$$

$$+ (-32) C_8(\mathbf{1,5,3,6}) C_9(\mathbf{2,5,4,6}) \delta_G(\mathbf{1+5+3+6}) \Big\} , \quad (34e)$$

$$\partial_\ell C_3\left(\mathbf{1,2,3,4}\right) = \delta_G\left(\mathbf{1+2-3-4}\right)\Big\{(-1)\,C_5\left(\mathbf{\text{-}3,1,2,4}\right)\Gamma\left(\mathbf{\text{-}3}\right)+(-1)\,C_5\left(\mathbf{\text{-}4,1,2,3}\right)\Gamma\left(\mathbf{\text{-}4}\right)$$

$$+(-1)\,C_7\left(\mathbf{\text{-}2,1,3,4}\right)\Gamma\left(\mathbf{\text{-}2}\right)+(-1)\,C_7\left(\mathbf{\text{-}1,2,3,4}\right)\Gamma\left(\mathbf{\text{-}1}\right)+$$

$$\sum_5 (-4)\,C_5\left(\mathbf{5,2,6,4}\right)C_7\left(\mathbf{5,1,3,6}\right)\delta_G\left(\mathbf{5+1+6-4}\right)+$$

$$(-4)\,C_5\left(\mathbf{5,1,6,3}\right)C_7\left(\mathbf{5,2,4,6}\right)\delta_G\left(\mathbf{5+1+6-3}\right)+$$

$$(-4)\,C_5\left(\mathbf{5,2,6,3}\right)C_7\left(\mathbf{5,1,4,6}\right)\delta_G\left(\mathbf{5+1+6-4}\right)+$$

$$(-4)\,C_5\left(\mathbf{5,1,6,4}\right)C_7\left(\mathbf{5,2,3,6}\right)\delta_G\left(\mathbf{5+1+6-3}\right)+$$

$$(-4)\,C_8\left(\mathbf{5,6,1,2}\right)C_9\left(\mathbf{5,6,3,4}\right)\delta_G\left(\mathbf{5+6+1+2}\right)\Big\}\,, \tag{34f}$$

$$\partial_\ell C_4\left(\mathbf{1,2,3,4}\right) = \delta_G\left(\mathbf{1+2-3+4}\right)\Big\{\left(\omega\left(\mathbf{3}\right)-\omega\left(\mathbf{1}\right)-\omega\left(\mathbf{2}\right)-\omega\left(\mathbf{4}\right)\right)C_4\left(\mathbf{1,2,3,4}\right)$$

$$+(-2)\,C_1\left(\mathbf{1,2,3,\text{-}4}\right)\Gamma\left(\mathbf{\text{-}4}\right)+(-4)\,C_8\left(\mathbf{1,2,4,\text{-}3}\right)\Gamma\left(\mathbf{3}\right)+$$

$$\left(-\frac{1}{2}\right)\left(C_2\left(\mathbf{2,3,4,\text{-}1}\right)\Gamma\left(\mathbf{1}\right)+C_2\left(\mathbf{1,3,4,\text{-}2}\right)\Gamma\left(\mathbf{2}\right)\right)+$$

$$\sum_{5,6}(-2)\,C_1\left(\mathbf{1,2,5,6}\right)C_4\left(\mathbf{5,6,3,4}\right)\delta_G\left(\mathbf{1+2-5-6}\right)+$$

$$(-1)\,C_2\left(\mathbf{2,5,4,6}\right)C_4\left(\mathbf{1,5,3,6}\right)\delta_G\left(\mathbf{2-5+4-6}\right)+$$

$$(-1)\,C_2\left(\mathbf{1,5,4,6}\right)C_4\left(\mathbf{2,5,3,6}\right)\delta_G\left(\mathbf{1-5+4-6}\right)+$$

$$(-8)\,C_6\left(\mathbf{2,3,5,6}\right)C_8\left(\mathbf{1,5,4,6}\right)\delta_G\left(\mathbf{2-3-5-6}\right)+$$

$$(-8)\,C_6\left(\mathbf{1,3,5,6}\right)C_8\left(\mathbf{2,5,4,6}\right)\delta_G\left(\mathbf{1-3-5-6}\right)+$$

$$(-4)\,C_7\left(\mathbf{3,4,5,6}\right)C_8\left(\mathbf{1,2,5,6}\right)\delta_G\left(\mathbf{3-4+5+6}\right)\Big\}\,, \tag{34g}$$

$$\partial_\ell C_5\left(\mathbf{1,2,3,4}\right) = \delta_G\left(\mathbf{1+2+3-4}\right)\Big\{\left(-\omega\left(\mathbf{1}\right)-\omega\left(\mathbf{2}\right)-\omega\left(\mathbf{3}\right)+\omega\left(\mathbf{4}\right)\right)C_5\left(\mathbf{1,2,3,4}\right)+$$

$$(-2)\,C_3\left(\mathbf{2,3,4,\text{-}1}\right)\Gamma\left(\mathbf{1}\right)+(-4)\,C_8\left(\mathbf{1,\text{-}4,2,3}\right)\Gamma\left(\mathbf{\text{-}4}\right)+$$

$$\left(-\frac{1}{2}\right)\left(C_2\left(\mathbf{1,\text{-}2,3,4}\right)\Gamma\left(\mathbf{\text{-}2}\right)+C_2\left(\mathbf{1,\text{-}3,2,4}\right)\Gamma\left(\mathbf{\text{-}3}\right)\right)+$$

$$\sum_{5,6}(-2)\,C_3\left(\mathbf{2,3,5,6}\right)C_5\left(\mathbf{1,5,6,4}\right)\delta_G\left(\mathbf{2+3-5-6}\right)+$$

$$(-1)\,C_2\left(\mathbf{1,5,3,6}\right)C_5\left(\mathbf{5,2,6,4}\right)\delta_G\left(\mathbf{1-5+3-6}\right)+$$

$$(-1)\,C_2\left(\mathbf{1,5,2,6}\right)C_5\left(\mathbf{5,3,6,4}\right)\delta_G\left(\mathbf{1-5+2-6}\right)+$$

$$(-8)\,C_7\left(\mathbf{5,3,4,6}\right)C_8\left(\mathbf{1,5,2,6}\right)\delta_G\left(\mathbf{5-3+4+6}\right)+$$

$$(-8)\,C_7\left(\mathbf{5,2,4,6}\right)C_8\left(\mathbf{1,5,3,6}\right)\delta_G\left(\mathbf{5-2+4+6}\right)+$$

$$(-4)\,C_6\left(\mathbf{1,5,6,4}\right)C_8\left(\mathbf{5,6,2,3}\right)\delta_G\left(\mathbf{1-5-6-4}\right)\Big\}\,, \tag{34h}$$

$$\partial_\ell C_6(1,2,3,4) = \delta_G(1-2-3-4)\Big\{(\omega(1)-\omega(2)-\omega(3)-\omega(4))\,C_6(1,2,3,4)+$$
$$(-2)\,C_1(1,\text{-}4,2,3)\,\Gamma(\text{-}4)+(-4)\,C_9(2,3,4,\text{-}1)\,\Gamma(1)+$$
$$\left(-\frac{1}{2}\right)\Big(C_2(1,3,\text{-}2,4)\,\Gamma(2)+C_2(1,2,\text{-}3,4)\,\Gamma(3)\Big)+$$
$$\sum_{5,6}(-2)\,C_1(5,6,2,3)\,C_6(1,5,6,4)\,\delta_G(5+6-2-3)+$$
$$(-1)\,C_2(5,3,6,4)\,C_6(1,2,5,6)\,\delta_G(5-3+6-4)+$$
$$(-1)\,C_2(5,2,6,4)\,C_6(1,3,5,6)\,\delta_G(5-2+6-4)+$$
$$(-8)\,C_4(1,5,3,6)\,C_9(2,5,4,6)\,\delta_G(1+5-3+6)+$$
$$(-8)\,C_4(1,5,2,6)\,C_9(3,5,4,6)\,\delta_G(1+5-2+6)+$$
$$(-4)\,C_5(1,5,6,4)\,C_9(2,3,5,6)\,\delta_G(1+5+6-4)\Big\}, \tag{34i}$$

$$\partial_\ell C_7(1,2,3,4) = \delta_G(1-2+3+4)\Big\{(-\omega(1)+\omega(2)-\omega(3)-\omega(4))\,C_7(1,2,3,4)+$$
$$(-2)\,C_3(\text{-}1,2,3,4)\,\Gamma(1)+(-4)\,C_9(1,\text{-}2,3,4)\,\Gamma(2)+$$
$$\left(-\frac{1}{2}\right)\Big(C_2(\text{-}4,1,2,3)\,\Gamma(4)+C_2(\text{-}3,1,2,4)\,\Gamma(3)\Big)+$$
$$\sum_{5,6}(-2)\,C_3(5,6,3,4)\,C_7(1,2,5,6)\,\delta_G(5+6-3-4)+$$
$$(-1)\,C_2(5,1,6,4)\,C_7(5,2,3,6)\,\delta_G(5-1+6-4)+$$
$$(-1)\,C_2(5,1,6,3)\,C_7(5,2,4,6)\,\delta_G(5-1+6-3)+$$
$$(-8)\,C_5(5,2,6,4)\,C_9(1,5,3,6)\,\delta_G(5+2+6-4)+$$
$$(-8)\,C_5(5,2,6,3)\,C_9(1,5,4,6)\,\delta_G(5+2+6-3)+$$
$$(-4)\,C_4(5,6,1,2)\,C_9(5,6,3,4)\,\delta_G(5+6-1+2)\Big\}, \tag{34j}$$

$$\partial_\ell C_8(1,2,3,4) = \delta_G(1+2+3+4)\Big\{(-\omega(1)-\omega(2)-\omega(3)-\omega(4))\,C_8(1,2,3,4)+$$
$$\sum_{5,6}(-2)\,C_1(1,2,5,6)\,C_8(5,6,3,4)\,\delta_G(1+2-5-6)+$$
$$(-1)\,C_2(2,5,4,6)\,C_8(1,5,3,6)\,\delta_G(2-5+4-6)+$$
$$(-1)\,C_2(1,5,4,6)\,C_8(2,5,3,6)\,\delta_G(1-5+4-6)+$$
$$(-1)\,C_2(2,5,3,6)\,C_8(1,5,4,6)\,\delta_G(2-5+3-6)+$$
$$(-1)\,C_2(1,5,3,6)\,C_8(2,5,4,6)\,\delta_G(1-5+3-6)+$$
$$(-2)\,C_3(3,4,5,6)\,C_8(1,2,5,6)\,\delta_G(3+4-5-6)\Big\}, \tag{34k}$$

$$\partial_\ell C_9(1,2,3,4) = \delta_G(1+2+3+4) \Big\{ (-\omega(1) - \omega(2) - \omega(3) - \omega(4)) C_9(1,2,3,4) +$$

$$\sum_{5,6} (-2) C_1(5,6,1,2) C_9(5,6,3,4) \delta_G(5+6-1-2) +$$

$$(-1) C_2(5,2,6,4) C_9(1,5,3,6) \delta_G(5-2+6-4) +$$

$$(-1) C_2(5,1,6,4) C_9(2,5,3,6) \delta_G(5-1+6-4) +$$

$$(-1) C_2(5,2,6,3) C_9(1,5,4,6) \delta_G(5-2+6-3) +$$

$$(-1) C_2(5,1,6,3) C_9(2,5,4,6) \delta_G(5-1+6-3) +$$

$$(-2) C_3(5,6,3,4) C_9(1,2,5,6) \delta_G(3+4-5-6) \Big\} . \tag{34l}$$

### A.2.4  Truncation of observables

Similar to the Hamilton operator, the flowing observables have to be truncated as well. But the corresponding spin operators are not evaluated directly, but as part of a resolvent. Hence, we do not truncate the spin operators in terms of scaling dimension. Instead, we keep those terms which couple the ground state to the relevant magnon channels. As we are interested in the subspaces with up to three magnons, we include operator terms up to a cubic level in annihilation and creation operators. Then, the flowing observables read

$$S^z(\mathbf{Q}) = \sum_{1,2} \Big\{ s_1^z(\mathbf{Q},1,2) \alpha_1^\dagger \alpha_2 + s_2^z(\mathbf{Q},1,2) \beta_1^\dagger \beta_2 + s_3^z(\mathbf{Q},1,2) \alpha_1 \beta_2 + s_4^z(\mathbf{Q},1,2) \alpha_1^\dagger \beta_2^\dagger \Big\} + C(\mathbf{Q}) ,$$
$$\tag{35}$$

with initial conditions

$$s_1^z(\mathbf{Q},1,2)\big|_{\ell=0} = \delta_G(\mathbf{Q}-1+2) \big[ \Gamma_{\mathbf{Q}-1+2} l_1 l_2 - m_1 m_2 \big] , \tag{36a}$$

$$s_2^z(\mathbf{Q},1,2)\big|_{\ell=0} = \delta_G(\mathbf{Q}-1+2) \big[ \Gamma_{\mathbf{Q}-1+2} m_1 m_2 - l_1 l_2 \big] , \tag{36b}$$

$$s_3^z(\mathbf{Q},1,2)\big|_{\ell=0} = \delta_G(\mathbf{Q}+1+2) \big[ \Gamma_{\mathbf{Q}-1+2} m_1 l_2 - l_1 m_2 \big] , \tag{36c}$$

$$s_4^z(\mathbf{Q},1,2)\big|_{\ell=0} = \delta_G(\mathbf{Q}-1-2) \big[ \Gamma_{\mathbf{Q}-1+2} m_1 l_2 - m_1 l_2 \big] , \tag{36d}$$

$$C(\mathbf{Q})\big|_{\ell=0} = \big( m_Q^2 - SN \big) \big( \Gamma_\mathbf{Q} - 1 \big) \delta_G(\mathbf{Q}) , \tag{36e}$$

$$S^+(\mathbf{Q}) = \sum_1 s_1^+(\mathbf{Q}) 1 \alpha_1 + s_2^+(\mathbf{Q}) 1 \beta_1^\dagger + \sum_{1,2,3} \Big\{ s_3^+(\mathbf{Q},1,2,3) \beta_1^\dagger \beta_2^\dagger \beta_3$$

$$+ s_4^+(\mathbf{Q},1,2,3) \alpha_1^\dagger \alpha_2 \alpha_3 + s_5^+(\mathbf{Q},1,2,3) \alpha_1^\dagger \beta_2^\dagger \beta_3^\dagger + s_6^+(\mathbf{Q},1,2,3) \alpha_1 \beta_2^\dagger \beta_3$$

$$+ s_7^+(\mathbf{Q},1,2,3) \alpha_1^\dagger \alpha_2 \beta_3^\dagger + s_8^+(\mathbf{Q},1,2,3) \alpha_1 \alpha_2 \beta_3 \Big\} , \tag{37}$$

with initial conditions

$$s_1^+(\mathbf{Q})\big|_{\ell=0} = \sqrt{2SN}\left(1 - \frac{1}{SN}\sum_{\mathbf{k}} m_{\mathbf{k}}^2\right)(l_1 + \Gamma_1 m_1),\tag{38a}$$

$$s_2^+(\mathbf{Q})\big|_{\ell=0} = \sqrt{2SN}\left(1 - \frac{1}{SN}\sum_{\mathbf{k}} m_{\mathbf{k}}^2\right)(m_1 + \Gamma_1 l_1),\tag{38b}$$

$$s_3^+(\mathbf{Q},1,2,3)\big|_{\ell=0} = \delta_G(\mathbf{Q}-1-2+3)\left[m_1 m_2 m_3 + \Gamma_{\mathbf{Q}-1-2+3} l_1 l_2 l_3\right],\tag{38c}$$

$$s_4^+(\mathbf{Q},1,2,3)\big|_{\ell=0} = \delta_G(\mathbf{Q}-1+2+3)\left[l_1 l_2 l_3 + \Gamma_{\mathbf{Q}-1+2+3} m_1 m_2 m_3\right],\tag{38d}$$

$$s_5^+(\mathbf{Q},1,2,3)\big|_{\ell=0} = \delta_G(\mathbf{Q}-1-2-3)\left[l_1 m_2 m_3 + \Gamma_{\mathbf{Q}-1-2-3} m_1 m_2 l_3\right],\tag{38e}$$

$$s_6^+(\mathbf{Q},1,2,3)\big|_{\ell=0} = 2\,\delta_G(\mathbf{Q}+1-2+3)\left[l_1 l_2 l_3 + \Gamma_{\mathbf{Q}+1-2+3} m_1 m_2 m_3\right],\tag{38f}$$

$$s_7^+(\mathbf{Q},1,2,3)\big|_{\ell=0} = 2\,\delta_G(\mathbf{Q}-1+2-3)\left[l_1 l_2 l_3 + \Gamma_{\mathbf{Q}-1+2-3} m_1 m_2 m_3\right],\tag{38g}$$

$$s_8^+(\mathbf{Q},1,2,3)\big|_{\ell=0} = \delta_G(\mathbf{Q}+1+2+3)\left[l_1 l_2 l_3 + \Gamma_{\mathbf{Q}+1+2+3} m_1 m_2 m_3\right],\tag{38h}$$

and

$$S^-(\mathbf{Q}) = s_1^-(\mathbf{Q})\,\alpha_{\mathbf{Q}}^\dagger + s_2^-(-\mathbf{Q})\,\beta_{-\mathbf{Q}} + \sum_{1,2,3}\Big\{ s_3^-(\mathbf{Q},1,2,3)\,\alpha_1^\dagger \alpha_2^\dagger \beta_3^\dagger$$
$$+ s_4^-(\mathbf{Q},1,2,3)\,\alpha_1^\dagger \alpha_2^\dagger \alpha_3 + s_5^-(\mathbf{Q},1,2,3)\,\alpha_1^\dagger \beta_2^\dagger \beta_3 + s_6^-(\mathbf{Q},1,2,3)\,\alpha_1^\dagger \alpha_2 \beta_3$$
$$+ s_7^-(\mathbf{Q},1,2,3)\,\beta_1^\dagger \beta_2 \beta_3 + s_8^-(\mathbf{Q},1,2,3)\,\alpha_1 \beta_2 \beta_3 \Big\},\tag{39}$$

with initial conditions

$$s_1^-(\mathbf{Q})\big|_{\ell=0} = \sqrt{2SN}\left(l_{\mathbf{Q}} + \Gamma_{\mathbf{Q}} m_{\mathbf{Q}}\right),\tag{40a}$$

$$s_2^-(\mathbf{Q})\big|_{\ell=0} = \sqrt{2SN}\left(m_{\mathbf{Q}} + \Gamma_{\mathbf{Q}} l_{\mathbf{Q}}\right),\tag{40b}$$

$$s_3^-(\mathbf{Q},1,2,3)\big|_{\ell=0} = 0,\qquad s_4^-(\mathbf{Q},1,2,3)\big|_{\ell=0} = 0,\qquad s_5^-(\mathbf{Q},1,2,3)\big|_{\ell=0} = 0,\tag{40c}$$

$$s_6^-(\mathbf{Q},1,2,3)\big|_{\ell=0} = 0,\qquad s_7^-(\mathbf{Q},1,2,3)\big|_{\ell=0} = 0,\qquad s_8^-(\mathbf{Q},1,2,3)\big|_{\ell=0} = 0.\tag{40d}$$

The flow of $C(\mathbf{Q})$ does not influence the other coefficients of the observables and it is not required in the evaluation of the relevant resolvents. Thus, it is not considered in the flow equations.

### A.2.5 Flow equations: Observables

The flow equations of the observables read

$$\partial_\ell s_1^z(\mathbf{Q},1,2) = \delta_G(\mathbf{Q}+1-2)\Big\{(-1)\Gamma(1) s_3^z(\mathbf{Q},2,\text{-}1) + (-1)\Gamma(2) s_4^z(\mathbf{Q},1,\text{-}2)$$
$$+ \sum_{3,4}(-2)C_4(1,3,2,4) s_3^z(\mathbf{Q},3,4)\,\delta_G(1+3-2+4) +$$
$$(-2)C_6(1,2,3,4) s_4^z(\mathbf{Q},3,4)\,\delta_G(1-2-3-4)\Big\},\tag{41a}$$

$$\partial_\ell s_2^z(\mathbf{Q},1,2) = \delta_G(\mathbf{Q}+1-2)\Big\{(-1)\Gamma(1) s_4^z(\mathbf{Q},\text{-}2,1) + (-1)\Gamma(2) s_3^z(\mathbf{Q},\text{-}1,2)$$
$$+ \sum_{3,4}(-2)C_5(3,1,4,2) s_3^z(\mathbf{Q},3,4)\,\delta_G(3+1+4-2) +$$
$$(-2)C_7(3,1,2,4) s_4^z(\mathbf{Q},3,4)\,\delta_G(3-1+2+4)\Big\},\tag{41b}$$

$$\partial_\ell s_3^z(\mathbf{Q}, 1, 2) = \delta_G(\mathbf{Q} - 1 - 2)\Big\{(-1)\Gamma(\text{-}2)s_1^z(\mathbf{Q}, \text{-}2, 1) + (-1)\Gamma(1)s_2^z(\mathbf{Q}, \text{-}1, 2)$$
$$+ \sum_{3,4}(-4)C_9(1, 3, 2, 4)s_4^z(\mathbf{Q}, 3, 4)\delta_G(1 + 3 + 2 + 4)\Big\}, \tag{41c}$$

$$\partial_\ell s_4^z(\mathbf{Q}, 1, 2) = \delta_G(\mathbf{Q} + 1 + 2)\Big\{(-1)\Gamma(\text{-}2)s_1^z(\mathbf{Q}, 1, \text{-}2) + (-1)\Gamma(1)s_2^z(\mathbf{Q}, 2, \text{-}1)$$
$$+ \sum_{3,4}(-4)C_8(1, 3, 2, 4)s_4^z(\mathbf{Q}, 3, 4)\delta_G(1 + 3 + 2 + 4)\Big\}, \tag{41d}$$

$$\partial_\ell s_1^+(\mathbf{Q}) = (-1)\Gamma(\mathbf{Q})s_2^+(\text{-}\mathbf{Q}) + \sum_2\Big\{(-1)\Gamma(2)s_7^+(\mathbf{Q}, 2, \mathbf{Q}, \text{-}2) + (-2)\Gamma(2)s_8^+(\mathbf{Q}, \mathbf{Q}, 2, \text{-}2)\Big\}$$
$$+ \sum_{2,3,4}\Big\{(-2)C_4(2, 3, \mathbf{Q}, 4)s_8^+(\mathbf{Q}, 2, 3, 4)\delta_G(2 + 3 - \mathbf{Q} + 4)$$
$$+ (-4)C_9(\mathbf{Q}, 2, 3, 4)s_5^+(\mathbf{Q}, 2, 3, 4) + \delta_G(\mathbf{Q} + 2 + 3 + 4)\Big\}, \tag{42a}$$

$$\partial_\ell s_2^+(\mathbf{Q}) = -\Gamma(\text{-}\mathbf{Q})s_1^+(\text{-}\mathbf{Q}) + \sum_2\Big\{-2\Gamma(2)s_5^+(\text{-}, \mathbf{Q}, 2, \mathbf{Q})\text{-}2 - \Gamma(2)s_6^+(\text{-}\mathbf{Q}, 2, \mathbf{Q}, \text{-}2)\Big\}$$
$$+ \sum_{2,3,4}\Big\{(-2)C_7(2, \mathbf{Q}, 3, 4)s_5^+(\text{-}\mathbf{Q}, 2, 3, 4)\delta_G(2 - \mathbf{Q} + 3 + 4)$$
$$+ (-4)C_8(2, 3, \mathbf{Q}, 4)s_8^+(\text{-}\mathbf{Q}, 2, 3, 4)\delta_G(\mathbf{Q} + 2 + 3 + 4)\Big\}, \tag{42b}$$

$$\partial_\ell s_3^+(\mathbf{Q}, 1, 2, 3) = \delta_G(\mathbf{Q} + 1 + 2 - 3)\Big\{-C_5(\mathbf{Q}, 1, 2, 3)s_1^+(\mathbf{Q}) - \Gamma(\text{-}3)s_5^+(\text{-}\mathbf{Q}, \text{-}3, 1, 2)$$
$$+ \left(-\frac{1}{2}\right)\Gamma(\text{-}1)s_6^+(\mathbf{Q}, \text{-}1, 2, 3) + \left(-\frac{1}{2}\right)\Gamma(\text{-}2)s_6^+(\mathbf{Q}, \text{-}2, 1, 3)$$
$$+ \sum_{4,5}(-2)C_7(4, 1, 3, 5)s_5^+(\mathbf{Q}, 4, 2, 5)\delta_G(4 - 1 + 3 - 5) +$$
$$(-2)C_7(4, 1, 2, 5)s_5^+(\mathbf{Q}, 4, 3, 5)\delta_G(4 - 1 + 2 - 5) +$$
$$(-1)C_5(4, 1, 5, 3)s_6^+(\mathbf{Q}, 4, 1, 5)\delta_G(4 + 1 + 5 - 3) +$$
$$(-1)C_5(4, 2, 5, 3)s_6^+(\mathbf{Q}, 4, 2, 5)\delta_G(4 + 2 + 5 - 3) +$$
$$(-2)C_8(4, 5, 1, 2)s_8^+(\mathbf{Q}, 4, 5, 3)\delta_G(4 + 5 + 1 + 2)\Big\}, \tag{42c}$$

$$\partial_\ell s_4^+(\mathbf{Q}, 1, 2, 3) = \delta_G(\mathbf{Q} + 1 - 2 - 3)\Big\{-C_6(1, 2, 3, \text{-}\mathbf{Q})s_2^+(\text{-}\mathbf{Q}) - \Gamma(1)s_8^+(\mathbf{Q}, 2, 3, \text{-}1)$$
$$+ \left(-\frac{1}{2}\right)\Gamma(2)s_7^+(\mathbf{Q}, 1, 3, \text{-}2) + \left(-\frac{1}{2}\right)\Gamma(3)s_7^+(\mathbf{Q}, 1, 2, \text{-}3) +$$
$$\sum_{4,5}(-2)C_4(1, 4, 2, 5)s_8^+(\mathbf{Q}, 3, 4, 5)\delta_G(1 + 4 - 2 + 5) +$$
$$(-2)C_4(1, 4, 3, 5)s_8^+(\mathbf{Q}, 2, 4, 5)\delta_G(1 + 4 - 3 + 5) +$$
$$(-2)C_9(2, 3, 4, 5)s_5^+(\mathbf{Q}, 1, 4, 5)\delta_G(2 + 3 + 4 + 5) +$$
$$(-1)C_6(1, 2, 4, 5)s_7^+(\mathbf{Q}, 4, 3, 5)\delta_G(1 - 2 - 4 - 5) +$$
$$(-1)C_6(1, 3, 4, 5)s_7^+(\mathbf{Q}, 4, 2, 5)\delta_G(1 - 3 - 4 - 5)\Big\}, \tag{42d}$$

$$\partial_\ell s_5^+(\mathbf{Q},1,2,3) = \delta_G(\mathbf{Q}+1+2+3)\Big\{ C_5(1,2,3,\text{-}\mathbf{Q})s_2^+(\text{-}\mathbf{Q}) - 2C_8(1,\mathbf{Q},2,3)s_1^+(\mathbf{Q})$$

$$-\Gamma(1)s_3^+(\mathbf{Q},2,3,\text{-}1) - \left(\frac{1}{2}\right)[\Gamma(\text{-}2)s_7^+(\mathbf{Q},1,\text{-}2,3) + \Gamma(\text{-}3)s_7^+(\mathbf{Q},1,\text{-}3,2)]$$

$$+\sum_{4,5}(-2)C_8(4,5,2,3)s_4^+(\mathbf{Q},1,4,5)\delta_G(4+5+2+3)+$$

$$(-2)C_8(1,4,2,5)s_6^+(\mathbf{Q},4,3,5)\delta_G(1+4+2+5)+$$

$$(-2)C_8(1,4,3,5)s_6^+(\mathbf{Q},4,2,5)\delta_G(1+4+3+5)\Big\},\qquad(42\text{e})$$

$$\partial_\ell s_6^+(\mathbf{Q},1,2,3) = \delta_G(\mathbf{Q}-1+2-3)\Big\{ -2C_7(1,2,3,\text{-}\mathbf{Q})s_2^+(\text{-}\mathbf{Q}) - 2\Gamma(\text{-}2)s_8^+(\mathbf{Q},1,\text{-}2,3)$$

$$+(-2)\Gamma(1)s_3^+(\mathbf{Q},2,\text{-}1,3) + (-1)\Gamma(\text{-}3)s_7^+(\mathbf{Q},\text{-}3,1,2)+$$

$$\sum_{4,5}(-8)C_9(1,4,3,5)s_5^+(\mathbf{Q},4,2,5)\delta_G(1+4+3+5)+$$

$$(-4)C_5(1,4,2,5)s_8^+(\mathbf{Q},1,4,5)\delta_G(1+4+2-5)+$$

$$(-2)C_7(1,2,4,5)s_3^+(\mathbf{Q},4,5,3)\delta_G(1-2+4+5)+$$

$$(-2)C_7(4,2,3,5)s_7^+(\mathbf{Q},4,1,5)\delta_G(4-2+3+5)+$$

$$(-2)C_4(4,5,1,2)s_8^+(\mathbf{Q},4,5,3)\delta_G(4+5-1+2)\Big\},\qquad(42\text{f})$$

$$\partial_\ell s_7^+(\mathbf{Q},1,2,3) = \delta_G(\mathbf{Q}+1-2+3)\Big\{ -2C_4(1,\mathbf{Q},2,3)s_1^+(\mathbf{Q}) - 2\Gamma(\text{-}3)s_4^+(\mathbf{Q},1,2,\text{-}3)$$

$$-2\Gamma(2)s_5^+(\mathbf{Q},1,3,\text{-}2) + (-1)\Gamma(1)s_6^+(\mathbf{Q},2,3,\text{-}1)$$

$$+\sum_{4,5}(-8)C_8(1,4,3,5)s_8^+(\mathbf{Q},2,4,5)\delta_G(1+4+3+5)+$$

$$(-4)C_6(1,2,4,5)s_5^+(\mathbf{Q},4,3,5)\delta_G(1-2-4-5)+$$

$$(-2)C_4(4,5,3,3)s_4^+(\mathbf{Q},1,4,5)\delta_G(4+5-3+3)+$$

$$(-2)C_7(2,3,4,5)s_5^+(\mathbf{Q},1,4,5)\delta_G(2-3+4+5)+$$

$$(-2)C_4(1,4,2,5)s_6^+(\mathbf{Q},4,3,5)\delta_G(1+4-2+5)\Big\},\qquad(42\text{g})$$

$$\partial_\ell s_8^+(\mathbf{Q},1,2,3) = \delta_G(\mathbf{Q}-1-2-3)\Big\{ C_6(\mathbf{Q},1,2,3)s_1^+(\mathbf{Q}) - 2C_9(1,2,3,\text{-}\mathbf{Q})s_2^+(\text{-}\mathbf{Q})$$

$$-\Gamma(\text{-}3)s_4^+(\mathbf{Q},\text{-}3,1,2) - \left(\frac{1}{2}\right)[\Gamma(1)s_6^+(\mathbf{Q},2,\text{-}1,3) + \Gamma(2)s_6^+(\mathbf{Q},1,\text{-}2,3)]$$

$$+\sum_{4,5}(-2)C_9(1,2,4,5)s_3^+(\mathbf{Q},4,5,3)\delta_G(1+2+4+5)+$$

$$(-2)C_9(1,4,3,5)s_7^+(\mathbf{Q},4,2,5)\delta_G(1+4+3+5)+$$

$$(-2)C_9(2,4,3,5)s_7^+(\mathbf{Q},4,1,5)\delta_G(2+4+3+5)\Big\},\qquad(42\text{h})$$

$$\partial_\ell s_1^-(\mathbf{Q}) = -\Gamma(\mathbf{Q})s_2^-(\text{-}\mathbf{Q}) + \Big\{ -\Gamma(2)s_6^-(\text{-}\mathbf{Q},\mathbf{Q},2,\text{-}2) + -2\Gamma(2)s_3^-(\text{-}\mathbf{Q},\mathbf{Q},2,\text{-}2)\Big\}$$

$$+\sum_{2,3,4}\Big\{(-2)C_6(\mathbf{Q},2,3,4)s_3^-(\text{-}\mathbf{Q},2,3,4)\delta_G(\mathbf{Q}-2-3-4)$$

$$+(-4)C_8(\mathbf{Q},2,3,4)s_8^-(\mathbf{Q},2,3,4)\delta_G(\mathbf{Q}-2+3+4)\Big\},\qquad(43\text{a})$$

$$\partial_\ell s_2^-(\mathbf{Q}) = -\Gamma(\mathbf{Q})s_1^-(\mathbf{Q}) - 2\Gamma(2)s_8^-(\mathbf{Q},2,\mathbf{Q},\text{-}2) - \Gamma(2)s_5^-(\mathbf{Q},2,\text{-}2,\mathbf{Q})\Big\}$$
$$+ \sum_{2,3,4}\Big\{(-2)C_5(2,3,4,\mathbf{Q})s_8^-(\mathbf{Q},2,3,4)\delta_G(2+3+4-\mathbf{Q})$$
$$+ (-4)C_9(2,3,\mathbf{Q},4)s_3^-(\mathbf{Q},2,3,4)\delta_G(\mathbf{Q}+2+3+4)\Big\}, \tag{43b}$$

$$\partial_\ell s_3^-(\mathbf{Q},1,2,3) = \delta_G(\mathbf{Q}+1+2+3)\Big\{C_4(1,2,\text{-}\mathbf{Q},3)s_1^-(\text{-}\mathbf{Q}) - \Gamma(\text{-}3)s_4^-(\mathbf{Q},1,2,\mathbf{Q})$$
$$- \left(\frac{1}{2}\right)[\Gamma(1)s_5^-(\mathbf{Q},2,3,\text{-}1) + \Gamma(2)s_5^-(\mathbf{Q},1,3,\text{-}2)] - 2C_8(1,2,3,\mathbf{Q})s_2^-(\mathbf{Q})$$
$$+ \sum_{4,5}(-2)C_8(1,2,4,5)s_7^-(\mathbf{Q},3,4,5)\delta_G(1+2+4+5)$$
$$+ (-2)C_8(1,4,3,5)s_6^-(\mathbf{Q},2,4,5)\delta_G(1+3+4+5)$$
$$+ (-2)C_8(2,4,3,5)s_6^-(\mathbf{Q},1,4,5)\delta_G(2+3+4+5)\Big\}, \tag{43c}$$

$$\partial_\ell s_4^-(\mathbf{Q},1,2,3) = \delta_G(\mathbf{Q}+1+2-3)\Big\{-C_5(1,2,3,\mathbf{Q})s_2^-(\mathbf{Q}) - \Gamma(3)s_3^-(\mathbf{Q},1,2,\text{-}3)$$
$$- \left(\frac{1}{2}\right)[\Gamma(1)s_6^-(\mathbf{Q},2,3,\text{-}1) + \Gamma(2)s_6^-(\mathbf{Q},1,3,\text{-}2)]$$
$$+ \sum_{4,5}(-1)C_4(1,4,3,5)s_6^-(\mathbf{Q},2,4,5)\delta_G(1+4-3+5)+$$
$$(-1)C_4(2,4,3,5)s_6^-(\mathbf{Q},1,4,5)\delta_G(2+4-3+5)+$$
$$(-2)C_6(1,3,4,5)s_3^-(\mathbf{Q},2,4,5)\delta_G(1-3-4-5)+$$
$$(-2)C_6(2,3,4,5)s_3^-(\mathbf{Q},1,4,5)\delta_G(2-3-4-5)+$$
$$(-2)C_8(1,2,4,5)s_8^-(\mathbf{Q},3,4,5)\delta_G(1+2+4+5)\Big\}, \tag{43d}$$

$$\partial_\ell s_5^-(\mathbf{Q},1,2,3) = \delta_G(\mathbf{Q}+1+2-3)\Big\{-2C_5(1,2,\mathbf{Q},3)s_2^-(\mathbf{Q}) - 2\Gamma(\text{-}3)s_3^-(\mathbf{Q},1,\text{-}3,2)$$
$$(-1)\Gamma(\text{-}2)s_6^-(\mathbf{Q},1,\text{-}2,3) + (-2)\Gamma(1)s_7^-(\mathbf{Q},2,3,\text{-}1)$$
$$+ \sum_{4,5}(-2)C_6(1,4,5,3)s_3^-(\mathbf{Q},4,5,2)\delta_G(1-4-5-3)+$$
$$(-4)C_7(4,2,3,5)s_3^-(\mathbf{Q},1,4,5)\delta_G(-4+2-3-5)+$$
$$(-2)C_5(4,2,5,3)s_6^-(\mathbf{Q},1,4,5)\delta_G(4+2+5-3)+$$
$$(-2)C_5(1,4,5,3)s_7^-(\mathbf{Q},2,4,5)\delta_G(1+4+5-3)+$$
$$(-8)C_8(1,4,2,5)s_8^-(\mathbf{Q},4,3,5)\delta_G(1+4+2+5)\Big\}, \tag{43e}$$

$$\partial_\ell s_6^-(\mathbf{Q},1,2,3) = \delta_G(\mathbf{Q}+1+2-3)\Big\{-2C_6(1,2,\text{-}\mathbf{Q},3)s_1^-(\text{-}\mathbf{Q}) - 2\Gamma(\text{-}3)s_4^-(\mathbf{Q},1,\text{-}3,2)$$
$$+ (-2)\Gamma(1)s_8^-(\mathbf{Q},2,3,\text{-}1) + (-1)\Gamma(2)s_5^-(\mathbf{Q},1,\text{-}2,3)$$
$$+ \sum_{4,5}(-2)C_6(1,4,5,3)s_4^-(\mathbf{Q},4,5,2)\delta_G(1-4-5-3)+$$
$$(-2)C_6(1,2,4,5)s_5^-(\mathbf{Q},4,5,3)\delta_G(1-2-4-5)+$$
$$(-2)C_5(1,4,5,3)s_8^-(\mathbf{Q},2,4,5)\delta_G(1+4+5-3)+$$
$$(-4)C_4(1,4,2,5)s_8^-(\mathbf{Q},4,3,5)\delta_G(1+4-2+5)+$$
$$(-8)C_9(2,4,3,5)s_3^-(\mathbf{Q},1,4,5)\delta_G(2+4+3+5)\Big\}, \tag{43f}$$

$$\partial_\ell s_7^- (\mathbf{Q}, 1, 2, 3) = \delta_G (\mathbf{Q} + 1 - 2 - 3) \Big\{ - C_8 (\text{-}\mathbf{Q}, 1, 2, 3) s_1^- (\text{-}\mathbf{Q}) - \Gamma(\text{-}1) s_8^- (\mathbf{Q}, \text{-}1, 2, 3)$$

$$- \left( \frac{1}{2} \right) [\Gamma(\text{-}2) s_5^- (\mathbf{Q}, \text{-}2, 1, 3) + \Gamma(\text{-}3) s_5^- (\mathbf{Q}, \text{-}3, 1, 2)]$$

$$+ \sum_{4,5} (-1) C_7 (4, 1, 2, 5) s_5^- (\mathbf{Q}, 4, 5, 3) \delta_G (4 - 1 + 2 + 5) +$$

$$(-1) C_7 (4, 1, 3, 5) s_5^- (\mathbf{Q}, 4, 5, 2) \delta_G (4 - 1 + 3 + 5) +$$

$$(-2) C_5 (4, 1, 5, 2) s_8^- (\mathbf{Q}, 4, 3, 5) \delta_G (4 + 1 - 5 + 2) +$$

$$(-2) C_5 (4, 1, 5, 3) s_8^- (\mathbf{Q}, 4, 2, 5) \delta_G (4 + 1 - 5 + 3) +$$

$$(-2) C_9 (4, 5, 2, 3) s_3^- (\mathbf{Q}, 4, 5, 1) \delta_G (2 + 4 + 3 + 5) \Big\} , \tag{43g}$$

$$\partial_\ell s_8^- (\mathbf{Q}, 1, 2, 3) = \delta_G (\mathbf{Q} - 1 - 2 - 3) \Big\{ C_7 (1, \mathbf{Q}, 2, 3) s_2^- (\mathbf{Q}) - 2 C_9 (1, \text{-}\mathbf{Q}, 2, 3) s_1^- (\text{-}\mathbf{Q}) -$$

$$\Gamma(1) s_7^- (\mathbf{Q}, \text{-}1, 2, 3) - \left( \frac{1}{2} \right) [\Gamma(\text{-}2) s_6^- (\mathbf{Q}, \text{-}2, 1, 3) + \Gamma(\text{-}3) s_6^- (\mathbf{Q}, \text{-}3, 1, 2)]$$

$$+ \sum_{4,5} (-2) C_9 (4, 5, 2, 3) s_4^- (\mathbf{Q}, 4, 5, 1) \delta_G (2 + 4 + 3 + 5) +$$

$$(-2) C_9 (1, 4, 2, 5) s_5^- (\mathbf{Q}, 4, 5, 3) \delta_G (1 + 4 + 2 + 5) +$$

$$(-2) C_9 (1, 4, 3, 5) s_5^- (\mathbf{Q}, 4, 5, 2) \delta_G (1 + 4 + 3 + 5) \Big\}. \tag{43h}$$

## B  Spectral densities

The general description of spectral densities is given in the main text in Sect. II.B. Explicit formulae are given in the sequel.

### B.1  Transversal dynamic structure factor

The transversal dynamic structure factor is given by

$$S^{xx+yy}(\omega, \mathbf{Q}) = -\frac{1}{\pi} \text{Im} \left( \langle 0 | S_{\text{eff}}^- (-\mathbf{Q}) \frac{1}{\omega - (H_{\text{eff}} - \bar{E}_0)} S_{\text{eff}}^+ (\mathbf{Q}) | 0 \rangle \right). \tag{44}$$

It splits into a one-magnon contribution

$$S^{xx+yy}(\omega, \mathbf{Q}) \big|_{1\text{mag}} = -\frac{1}{\pi} \text{Im} \left( \langle 0 | s_2^- (-\mathbf{Q}) \beta_{\mathbf{Q}} \frac{1}{\omega - (H_{\text{eff}} - \bar{E}_0)} s_2^+ (\mathbf{Q}) \beta_{\mathbf{Q}}^\dagger | 0 \rangle \right) \tag{45a}$$

$$= s_2^- (-\mathbf{Q}) s_2^+ (\mathbf{Q}) \delta (\omega - \omega(\mathbf{Q})) , \tag{45b}$$

where $W^{1\text{mag}} = s_2^- (-\mathbf{Q}) s_2^+ (\mathbf{Q})$ defines the one-magnon spectral weight, and a three-magnon contribution

$$S^{xx+yy}(\omega, \mathbf{Q}) \big|_{3\text{mag}} =$$

$$\sum_{1,2} \langle 0 | \alpha_2 \beta_{\mathbf{Q}-2} s_3^z (-\mathbf{Q}, 2, \mathbf{Q} - 2) \frac{1}{\omega - (H_{\text{eff}} - \bar{E}_0)} s_4^z (\mathbf{Q}, 1, \mathbf{Q} - 1) \alpha_1^\dagger \beta_{\mathbf{Q}-1}^\dagger | 0 \rangle , \tag{46}$$

which constitutes the incoherent part of the spectral density. The transversal static structure factor $W^{3-\text{mag}}(\mathbf{Q}) = \sum_1 s_3^z (-\mathbf{Q}, 1, \mathbf{Q} - 1) s_4^z (\mathbf{Q}, 1, \mathbf{Q} - 1)$ provides the total weight in the spectral density $S^{xx+yy}(\omega, \mathbf{Q}) \big|_{3\text{mag}}$, i.e., in the three-magnon channel. The static structure factor in the thermodynamic limit is obtained by an extrapolation in $\frac{1}{L} \to 0$.

## B.2 Longitudinal dynamic structure factor

The longitudinal dynamic structure factor is given by

$$S^{zz}(\omega, \mathbf{Q}) = \langle 0|S^z_{\text{eff}}(-\mathbf{Q}) \frac{1}{\omega - (H_{\text{eff}} - \bar{E}_0)} S^z_{\text{eff}}(\mathbf{Q})|0\rangle \tag{47a}$$

$$= \sum_{1,2} \langle 0|\alpha_2 \beta_{\mathbf{Q}-2} s^z_3 (-\mathbf{Q}, 2, \mathbf{Q}-2) \frac{1}{\omega - (H_{\text{eff}} - \bar{E}_0)} s^z_4 (\mathbf{Q}, 1, \mathbf{Q}-1) \alpha^\dagger_1 \beta^\dagger_{\mathbf{Q}-1}|0\rangle . \tag{47b}$$

Accordingly, its total weight is determined by the longitudinal static structure factor

$$W^{2-\text{mag}}(\mathbf{Q}) = \sum_{12} s^z_3(-\mathbf{Q}, 1, \mathbf{Q}-1) s^z_4(\mathbf{Q}, 1, \mathbf{Q}-1) . \tag{48}$$

The static structure factor in the thermodynamic limit is obtained by a linear extrapolation in $\frac{1}{L} \to 0$.

## B.3 Spectral weights

Integrating the spectral densities over frequency yields the spectral weights. They can be classified according to the number $n$ of magnons. The transversal weight is denoted by $S^{xx}_n(\mathbf{Q}) + S^{yy}_n(\mathbf{Q})$ and the longitudinal weight by $S^{zz}_n(\mathbf{Q})$. The extrapolated values for the thermodynamic limit are given in Tab. 1

| $Q_x$ | $Q_y$ | $S^{xx}_1 + S^{yy}_1$ | $S^{xx}_3 + S^{yy}_3$ | relative | $S^{zz}_2$ |
|---|---|---|---|---|---|
| $\frac{\pi}{2}$ | $\frac{\pi}{2}$ | 0.5839 | 0.1337 | 0.8137 | 0.2508 |
| $\frac{\pi}{3}$ | $\frac{2\pi}{3}$ | 0.5456 | 0.1749 | 0.7573 | 0.2496 |
| $\frac{\pi}{6}$ | $\frac{5\pi}{6}$ | 0.4705 | 0.2557 | 0.6479 | 0.2470 |
| $0$ | $\pi$ | 0.4339 | 0.2952 | 0.5951 | 0.2457 |
| $\frac{\pi}{3}$ | $\pi$ | 0.6575 | 0.2788 | 0.7022 | 0.3234 |
| $\frac{2\pi}{3}$ | $\pi$ | 1.5146 | 0.3313 | 0.8205 | 0.6177 |
| $\pi$ | $\pi$ | - | - | - | 1.1063 |
| $\frac{5\pi}{6}$ | $\frac{5\pi}{6}$ | 2.1826 | 0.3508 | 0.8615 | 0.8002 |
| $\frac{2\pi}{3}$ | $\frac{2\pi}{3}$ | 0.9576 | 0.2759 | 0.7763 | 0.4314 |
| $\frac{\pi}{2}$ | $\frac{\pi}{2}$ | 0.5839 | 0.1337 | 0.8137 | 0.2508 |
| $\frac{\pi}{3}$ | $\frac{\pi}{3}$ | 0.3435 | 0.0620 | 0.8471 | 0.1300 |
| $\frac{\pi}{6}$ | $\frac{\pi}{6}$ | 0.1603 | 0.0190 | 0.8941 | 0.0402 |
| $0$ | $0$ | 0 | 0 | 0 | 0 |
| $0$ | $\frac{\pi}{3}$ | 0.2235 | 0.0351 | 0.8643 | 0.0707 |
| $0$ | $\frac{2\pi}{3}$ | 0.3997 | 0.1570 | 0.7180 | 0.1853 |
| $0$ | $\pi$ | 0.4339 | 0.2952 | 0.5951 | 0.2457 |

Table 1: Spectral weights extrapolated to infinite system size for various channels at various high-symmetry points in the MBZ. The transversal channel comprises the one-magnon and the three-magnon channel. The third column shows the relative weight of the one-magnon channel, i.e., the ratio $(S^{xx}_1 + S^{yy}_1)/(S^{xx}_1 + S^{yy}_1 + S^{xx}_3 + S^{yy}_3)$. At $\mathbf{Q} = (0, 0)$ the DSF are identical zero because the corresponding operator does not introduce any excitation. At $\mathbf{Q} = (\pi, \pi)$ the static transversal DSF diverges so that no value can be given. The last column displays the two-magnon weight in the longitudinal channel. The lattice constant is set to unity.

### B.4  Non-symmetric continued fraction representation

The spectral densities in the multi-magnon subspaces are evaluated using a continued fraction representation for the resolvent [59]. Here, we are dealing with a non-symmetric problem

$$R(\omega) = \langle v_L | \frac{1}{\omega - H} | v_R \rangle \,, \tag{49}$$

with $H \neq H^\dagger$ and $\langle v_L | \neq \langle v_R |$ which leads to a generalized continued fraction representation

$$R(\omega) = \cfrac{\langle v_L | v_R \rangle}{\omega - a_0 - \cfrac{b_1 g_1}{\omega - a_1 - \cfrac{b_2 g_2}{\cdots}}} \,, \tag{50}$$

where we omitted any further indices on the magnon sector or the momentum for the sake of clarity. The coefficients $a_i$, $b_i$ and $g_i$ can be obtained by a non-symmetric Lanczos tridiagonalization [39] for $H$ with starting vectors $\langle v_L |$ and $| v_R \rangle$ yielding the tridiagonal matrix

$$H_{\text{tri}} = \begin{bmatrix} a_0 & b_1 & 0 & \cdots & 0 \\ g_1 & a_1 & b_2 & \ddots & \vdots \\ 0 & g_2 & a_2 & \ddots & 0 \\ \vdots & \ddots & \ddots & \ddots & b_m \\ 0 & \cdots & 0 & g_m & a_m \end{bmatrix} \,, \tag{51}$$

where $m$ is the number of Lanczos steps. The corresponding spectral density is a sum of weighted $\delta$-peaks located at the eigenvalues $\omega_i$ of $H_{\text{tri}}$

$$I(\omega) = -\frac{1}{\pi} \operatorname{Im} R(\omega) = \sum_{i=0}^{m} W_i \delta(\omega - \omega_i) \,. \tag{52}$$

which becomes a continuous function in the limit $m \to \infty$.

The weights $W_i$ are given by the overlap between the starting vectors and the corresponding left and right eigenvector of $H_{\text{tri}}$, i.e., $W_i = \langle L_i | v_R \rangle \langle v_L | R_i \rangle$ with $\langle L_i | H = \langle L_i | \omega_i$ and $H | R_i \rangle = \omega_i | R_i \rangle$. Note, that the eigenbasis is bi-orthonormal, meaning that $\langle L_i | R_j \rangle = \delta_{ij}$.

The asymptotic behavior of coefficients is related to the upper bound $E_u$ and lower bound $E_l$ of the continuum

$$E_u = a_\infty + 2\sqrt{g_\infty b_\infty} \,, \tag{53a}$$

$$E_l = a_\infty - 2\sqrt{g_\infty b_\infty} \,, \tag{53b}$$

see for instance Ref. [60]. Hence, it is possible to approximate the coefficients $a_m$ and $g_m b_m$ by $a_\infty$ and $g_\infty b_\infty$ for sufficiently large $m$ in the thermodynamic limit. In the case of a finite Hilbert space the behavior of coefficients will deviate from this asymptotic rule at larger $m$ depending on the actual size of the Hilbert space. To keep these finite-size effects minimum, we enlarge the Hilbert space by appropriate interpolation schemes increasing the number of grid points in the MBZ. This increases the maximum number $m_{\text{max}}$ of Lanczos steps before the coefficients start to deviate sizeably from the asymptotic behavior. For $m > m_{\text{max}}$ the sequence of coefficients is approximated by constant coefficients $a_\infty$ and $g_\infty b_\infty$. In generic recursion approaches, this infinite sequence in the continued fraction representation is equivalent to appropriate terminating functions [59, 60].

In order to reproduce the finite resolution in the experiment, we replace the $\delta$-peaks resulting from the diagonalization of the finite tridiagonal matrix $H_{\text{tri}}$ by Gaussian distribution functions with the corresponding weight $W_i$ and an artificial resolution which fits the broadening in the experimental data

$$\sum_i W_i \delta(\omega - \omega_i) \to I(\omega) = \sum_i W_i \frac{1}{\sqrt{2\sigma^2\pi}} e^{\frac{1}{2}\left(\frac{\omega-\omega_i}{\sigma}\right)^2} . \tag{54}$$

The values of $W_i$ and $\omega_i$ are found from the diagonalization of the interpolated system and the overlap with its eigen vectors. We normalize the broadened spectral functions $(\sum_i W_i)^{-1} I(\omega)$ and rescale them by the corresponding spectral weights in the thermodynamic limit. The latter are obtained by an extrapolation of the static structure factors in $1/L \to 0$, see Tab. 1. Thus, the continua in Fig. 4 in the main text are finally plotted according to

$$I_{\text{plot}}^{xx+yy}(\omega) = \frac{S_3^{xx}(\mathbf{Q}) + S_3^{yy}(\mathbf{Q})}{\sum_i W_i^{xx+yy}} I^{xx+yy}(\omega) , \tag{55a}$$

$$I_{\text{plot}}^{zz}(\omega) = \frac{S_2^{zz}(\mathbf{Q})}{\sum_i W_i^{zz}} I^{zz}(\omega) , \tag{55b}$$

in the transversal and in the longitudinal channel, respectively. This way, we avoid any bias of the relative total weights due to different interpolation schemes applied for different magnon sectors.

### B.5 Interpolation schemes

#### B.5.1 Dispersion

For the dispersion $\omega(\mathbf{k})$ we are using a Lanczos re-sampling method [61] which is suitable for the interpolation of periodic functions. In order to treat the linear cusp at $\mathbf{k} = 0$ appropriately we interpolate the function $f(\mathbf{k}) = \omega(\mathbf{k})^2$ instead, taking the square root afterwards.

#### B.5.2 Observables

The coefficient functions in the longitudinal observable depend on a two-dimensional wavevector like the dispersion. But these functions exhibit poles at certain momenta which makes it more complicated to interpolate them directly. In order to overcome this problem, we interpolate the function $f(\mathbf{k}) = \frac{\bar{s}(\mathbf{k})}{s(\mathbf{k})+c}$ where $\bar{s}(\mathbf{k})$ is the function transformed by the CST and $s(\mathbf{k})$ denotes the corresponding function *before* the CST which is known analytically. As the renormalization of the CST does not shift or alter the poles in $\bar{s}(\mathbf{k})$ the function $f(\mathbf{k})$ is finite and smooth. The constant shift $c$ is introduced to avoid zeroes in the denominator. This requires to track sign changes in $s(\mathbf{k})$ due to umklapp processes induced by the corresponding factors $\Gamma(\mathbf{K})$. Typically, we set the shift to $c = \alpha \cdot \min(s(k))$ with $\alpha \approx 1.5$

The coefficient functions in the three-magnon contribution of the transversal observable are directly interpolated using a quadrilinear interpolation scheme. Note, that the interpolation scheme used in the longitudinal part cannot be applied here because the functions in (39) before the CST are identical to zero. We stress that the three-magnon response is *only* induced by the renormalization in the course of the CST. The interpolation for the three-magnon contributions does not need to be as sophisticated as for the two-magnon contribution because the required number of intermediate grid points is much smaller compared to the longitudinal case. This is so because at fixed total momentum $\mathbf{Q}$ the dimension of the Hilbert space in the two-magnon subsector is $L^2$ while it is $L^4$ in the three-magnon sector.

### B.5.3 Interaction

The vertex functions with their six-dimensional arguments are interpolated directly by a multilinear or nearest neighbor interpolation scheme depending on the required number of grid points. The arguments of the vertex functions $C_i(\mathbf{k}_1, \mathbf{k}_2, \mathbf{k}_3, \mathbf{k}_4)$ are momenta defined in the first MBZ. Thus, the reciprocal vector $\Gamma = \sum_i k_i$ determined by the total momentum conservation may switch inside the first MBZ leading to a sign change in the coefficients. In order to avoid such jumps we track sign changes in the interpolation scheme.

## B.6 General procedure

In the following, we summarize the general procedure to calculate the continuous spectral densities briefly:

1. First, we perform a linear extrapolation of the effective coefficients at $L = 8$ and $L = 16$ in $\frac{1}{L}$ to $L = \infty$. We obtain an extrapolated effective model defined at $N = 8 \times 8$ grid points in the first MBZ. The same extrapolation scheme is applied to the static structure factors yielding the total weights of the two- and three-magnon contributions.

2. We interpolate the effective coefficients in order to increase the Hilbert space for the subsequent recursion analysis to reduce finite-size effects For the longitudinal densities, the system size is enhanced from $L = 8$ to $L = 192$. For the transversal three-magnon continuum, a system size of $L = 16$ is sufficient.

3. A non-symmetric Lanczos algorithm is applied to determine the continued fraction representation of the spectral densities. The list of the resulting coefficients is extended further by additional coefficients known from the asymptotic limit. We approximate the spectral densities by a sum of Gaussian distribution functions with uniform broadening. Their locations and weights are obtained by an exact diagonalization of the extended tridiagonal matrix.

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
