# Peer review of "Mutually attracting spin waves in the square-lattice quantum antiferromagnet"

_SciPost Physics, doi:SciPost Phys. 4, 001 (2018)_

## Round 2 · Referee Report · Anonymous (Referee 1) · 2017-11-30

Strengths

1 - addresses a fundamental model of quantum magnetism, in particular the nature of the excitations beyond the low-energy regime
2 - provides a quantitative account for the the spin dynamics as measured in neutron scattering within the interacting magnon picture
3 - direct comparison to recent experimental work performed
4 - well motivated, clear exposition, and overall clearly written text

Weaknesses

only minor, formal points - see report

Report

In their manuscript Powalski, Schmidt and Uhrig present a detailed and quantitative analysis of the magon dispersion and the dynamical spectral function of the spin-1/2 Heisenberg model on the square lattice. While this system has been studied in much detail in the past, there has recently been renewed interest in this basic model of quantum magnetism. This concerns in particular the nature of the excitations beyond the low energy regime, for two main reasons: (i) there are systematic differences in the magnon dispersion at short wavelengths between both inelastic neutron scatting data on high quality samples and numerical findings of the Heisenberg model with respect to the theoretical description of the magnon dispersion within perturbative expansions in 1/S, such as extended spin wave theory. (ii) These differences were suggested recently to be in agreement with a scenario with a fractionalization of magnons into spinons in the corresponding energy range, which called the conventional magnon picture into question.

The authors here provide a substantial extension of their previous study of the excitations in the spin-1/2 Heisenberg model, based on non-perturbative continuous similarity transformations. Taking into account the transformation of the measurement operators (the authors refer to these corrections as vertex corrections), they account for the continuum contributions in addition to the single magnon excitations, in particular for the transverse scattering channel. The authors show that such an advanced treatment can indeed reproduce the experimentally observed features in the neutron scattering data to a remarkably high degree. In particular, they argue that the observed dispersions as well as the enhanced continuum contribution to the scattering signal closely atop the single magnon peak in the elevated energy range result from magnon-magnon interactions and the hybridization with the three-magnon continuum. The authors systematically verify both effects by examining separately the various contributions to the effective Hamiltoniam within the magnon picture.

The remarkable agreement with the experimental data demonstrates that the magnon picture does indeed account for the spin dynamics of the Heisenberg model on the square lattice without the need to resort to a possible fractionalization of the magnons into spinons. In my opinion, this is a very important result in view of both recent developements and the significant amount of previous work on this very fundamental model of quantum magnetism.

The paper is well written, and several useful details are conveniently provided in the appendices.
I would still ask the the authors to give the paper one more proofreading. This does not only concern the text, but there are also
apparent mistakes in the questions. An easy one to spot is the missing imaginary unit in Eq. 3, but I leave it to the authors
to ensure that, e.g., the indices in their flow-equations have been checked carefully. Furthermore,
(i) Ref. 16 should be cited at the end of the last paragraph that ends on page 3 ("Whe show that...experimental results").
(ii) On page 16, correct "this physics is also be found...".
(iii) In the caption of Fig. 6 replace "The upper diagram is ..."->"The upper left diagram is ..." and "additional order" -> "additional orders"

Requested changes

see the few items listed in the report

---

## Editorial Decision

published